# `carps`: A Framework for Comparing N Hyperparameter Optimizers on M Benchmarks

**Carolin Benjamins**[1], **Helena Graf**[1], **Sarah Segel**[1], **Difan Deng**[1], **Tim Ruhkopf**[1] **Leona Hennig**[1],
**Soham Basu**[2], **Neeratyoy Mallik**[2], **Edward Bergman**[2] **Deyao Chen**[3], **François Clément**[4],
**Alexander Tornede**[1], **Matthias Feurer**[5,9], **Katharina Eggensperger**[6]
**Frank Hutter**[7,2], **Carola Doerr**[8], **Marius Lindauer**[1,10]
[1] *Leibniz University Hannover,* [2] *Albert-Ludwigs University Freiburg,* [3] *University of Oxford,* [4] *University of Washington,* [5] *LMU Munich,* [6] *University of Tübingen,* [7] *ELLIS Institute Tübingen,* [8] *Sorbonne University, CNRS,* [9] *Munich Center for Machine Learning,* [10] *L3S Research Center,*
*email Corresponding author email:* `c.benjamins@ai.uni-hannover.de`

**Reviewed on OpenReview:** *https://openreview.net/forum?id=AuA8m4I6zI*

## Abstract

Hyperparameter Optimization (HPO) is crucial to developing well-performing machine learning models. In order to ease prototyping and benchmarking of HPO methods, we propose `carps`, a benchmark framework for **C**omprehensive **A**utomated **R**esearch **P**erformance **S**tudies allowing to evaluate $N$ optimizers on $M$ benchmark tasks. In this first release of `carps`, we focus on the four most important types of HPO task types: blackbox, multi-fidelity, multi-objective, and multi-fidelity-multi-objective. With 3 336 tasks from 5 community benchmark collections and 26 variants of 9 optimizer families, we offer the largest go-to library to date for evaluating and comparing HPO methods. The `carps` framework relies on a purpose-built, lightweight interface that glues together optimizers and benchmark tasks. It also includes an analysis pipeline that facilitates the evaluation of optimizers on benchmarks. However, navigating a huge number of tasks while developing and comparing methods can be computationally infeasible. To address this, we obtain a subset of representative tasks by minimizing the star discrepancy of the subset in the space spanned by the full set. As a result, we propose an initial subset of 10 to 30 diverse tasks for each task type, and include functionality to recompute subsets as more benchmarks become available, enabling efficient evaluations. We also establish a first set of baseline results on these tasks as a measure for future comparisons. With `carps` (https://www.github.com/automl/CARP-S), we make an important step in the standardization of HPO evaluation.

## 1 Introduction

Hyperparameter optimization (HPO) is an integral step for improving the performance of machine learning (ML) methods (Feurer and Hutter, 2019; Bischl et al., 2023). In turn, to develop effective and reliable HPO methods, thorough benchmarking is necessary (Bischl et al., 2023), which has led to an increasing amount of benchmarks being developed for a variety of use cases (Eggensperger et al., 2015; Turner and Eriksson, 2019; Pineda Arango et al., 2021; Eggensperger et al., 2021; Pfisterer et al., 2022; Salinas et al., 2022). However, this vast number of tasks ironically impedes the benchmarking process, on the one hand, due to computational requirements and, on the other, due to potential bias from overrepresented task types. Furthermore, developers often go for the easiest-to-use benchmark package and not necessarily for the most informative one, particularly because the latter is not obvious. This poses three main challenges for benchmarking a new optimizer: (i) selecting appropriate benchmark tasks and baselines, (ii) including those different benchmark tasks and baselines into an experiment setup on a technical level, and (iii) often having to set up a new benchmarking setup from scratch, leading to a higher chance of reproducibility issues. To

address these challenges, we propose `carps`, an HPO benchmarking framework for **C**omprehensive **A**utomated **R**esearch **P**erformance **S**tudies, which allows evaluating $N$ optimizers on $M$ benchmarks.

`carps` provides a unified access to many benchmark collections and optimizers through a lightweight interface and offers a subselection of benchmarking tasks for targeted development of new HPO approaches. The initial version contains a diverse set of 9 optimizer families (see Table 3 in the appendix for an overview) and 5 benchmark collections (see Section 6), readily available. The lightweight interface between optimizer and benchmark eases the integration of new components, making the framework and one's own experiments easily extensible and scalable. The included benchmarks target different HPO task types, namely blackbox (BB), multi-fidelity (MF), multi-objective (MO), and multi-fidelity-multi-objective (MOMF). To ensure that new HPO approaches are not overengineered to a few HPO tasks, we propose separate development and test subsets of benchmarks for these task types. By minimizing the star discrepancy (Clément et al., 2024), we design these subsets to fully cover the entire task space. This allows to efficiently run representative benchmarks without going for the full collection of 3 336 tasks. In summary, `carps` eases the rapid development of optimizers and ensures quality benchmarking.

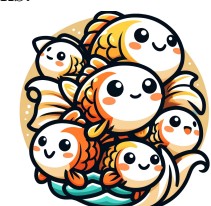

Figure 1: Our benchmarking framework `carps` eases developing new optimizers by providing a multitude of diverse benchmarks with a unified interface.

Our contributions are:

1. A comprehensive experimentation pipeline from defining and running HPO experiments to logging and analysis;

2. A lightweight and unified interface glueing optimizer and benchmark together for ease of integration (Section 4.1);

3. Easy access to 3 336 HPO tasks from 5 benchmark collections and 26 variants of 9 optimizers, with seamless deployment facilitated by Hydra, which supports launching their combinations on SLURM, Ray, RQ, and Joblib;

4. Disjoint benchmark subsets for development and testing each for blackbox (BB), multi-fidelity (MF), multi-objective (MO), and multi-fidelity-multi-objective (MOMF) optimization using the star discrepancy, allowing efficient evaluation of HPO optimizers (Section 7); as well as the functionality to recompute subsets as more benchmarks become available

5. Experimental results available for baseline methods on these benchmark tasks (Section 8).

## 2 Background: Hyperparameter Optimization

Before laying out `carps` as a framework with its optimizers and tasks and describing the subselection, we introduce the optimization setting and common terms.

### 2.1 Optimization: Setting and Terms

In optimization, we aim to optimize the *objective function*, which is a mapping from an *input space* $\mathcal{X}$ to an *output space* $\mathcal{O}$:

$$f : \mathcal{X} \to \mathcal{O}. \tag{1}$$

The output space is often half-bounded and has the number of objectives $O$ as dimensions: $\mathcal{O} = \{(y_0, \ldots, y_{O-1}) \in \mathbb{R}^O | \{y_i \geq y_{i,\min}\}_{i=0,\ldots,O-1}\}$. An objective dimension can also be fully bounded for an objective like accuracy $\mathrm{acc} \in [0, 1]$. The output space can also contain constraints (Garnett, 2023). The objective function can be of a stochastic nature (Garnett, 2023), where we seek to optimize the expected performance, e.g., for the physical process of baking a cake and finding the optimal oven hyperparameters. Next to that, the training of a neural network is also a stochastic process that becomes quasi-deterministic by defining a random state on the computer. When benchmarking, instead of querying the real objective

function, the objective function can also be replaced by cheap-to-evaluate alternatives like table lookups or surrogates (Eggensperger et al., 2015; 2019; Zela et al., 2022).

Before we elaborate on the input space and how it is composed, we define the *optimization setting*. Here, we aim to find the minimizer $x^*$ of the objective function[1]:

$$x^* \in \arg\min_{x \in \mathcal{X}} f(x) \tag{2}$$

In the case of a multi-dimensional output space, i.e., multiple objectives, $x^*$ is the Pareto front. The optimization setting will be refined after introducing the spaces that make up the input space.

The input space itself is a cartesian product of spaces, namely the *configuration space* $\Lambda$, *fidelity space* $\mathcal{F}$, and the *instance space* $\mathcal{I}$: $\mathcal{X} = \Lambda \times \mathcal{F} \times \mathcal{I}$. In addition, the objective function can also be of a stochastic nature.

The configuration space describes the classic parameters for the objective function. In HPO, the objective function is an algorithm to be optimized for a certain task. The parameters here are called *hyperparameters*. HPO can mean optimizing the hyperparameters of an ML model on a target dataset (Feurer and Hutter, 2019; Bischl et al., 2023) for different objectives like accuracy or runtime.

Hyperparameters can be of different types, and the most common are: Continuous (float, also on a log-scale), integers (int), ordinal (ord, ordered choices or ordered discrete numbers), and categorical (cat, unordered choices or elements). Next to different hyperparameter types, the configuration space can also contain constraints, conditions, and hierarchies. An example of a mixed configuration space with hierarchies would be the choice of different optimizers for a neural network, like Adam or SGD, where each choice of optimizer itself has its hyperparameters, which are different from the ones of the other optimizer choices.

In addition, for certain objective functions, we can query them using different levels of resources, called fidelity $F \in \mathcal{F}$, where we assume that querying the objective function $f$ with lower resources is an approximation of querying $f$ with full resources. An example would be the number of epochs of a training algorithm for a neural network. Instead of evaluating the full epochs, a hyperparameter configuration $\lambda \in \Lambda$ can only be evaluated with a small number of epochs to approximate the performance after a full training run. This principle is exploited in the area of multi-fidelity optimization (Jamieson and Talwalkar, 2016; Li et al., 2018), resulting in more resource-efficient algorithms. The instance space $\mathcal{I}$ becomes relevant in the field of algorithm configuration (Hutter et al., 2009), where we aim to find a well-performing hyperparameter configuration for a set of objective function instances.

Therefore, with this input space $\mathcal{X}$, performing one *trial* $\mathcal{T}$ means evaluating the objective function with a certain hyperparameter configuration $\lambda \in \Lambda$, fidelity $F \in \mathcal{F}$ and instance $I \in \mathcal{I}$, such that $x = (\lambda, F, I)$, and observing the objective function value: $T = (x, f(x))$. In general, we *only* aim to optimize in the configuration space, meaning optimizing the hyperparameters. The fidelity space and instance space belong to the input space of the objective function, but are not free parameters to be optimized. With this distinction, we define the *hyperparameter optimization setting* as finding the best hyperparameter configuration $\lambda^*$ at the highest fidelity $F_{\max}$ for one fixed instance $I_f$:

$$\lambda^* \in \arg\min_{\lambda \in \Lambda} f(\lambda, F = F_{\max}, I = I_f). \tag{3}$$

If the objective function has no notion of fidelity, it can also be viewed as always evaluating at the highest fidelity. The best performing hyperparameter configuration $\lambda^*$ is also called the *incumbent*.

One important aspect is missing when it comes to benchmarking and actual optimization. As described, the objective function receives the input and output space. However, it is possible to reduce the input and output space of the objective function. For example, we can only choose to optimize two of many possible hyperparameters and only one objective of many. Therefore, we define the *task* to be an objective function with an associated (sub) input space and (sub) output space and a computational budget, e.g., a fixed number of trials or how often we can query the objective function. Thus, we can have several tasks with the same underlying objective function. See Figure 2 for a schematic representation of a task.

---

[1]Note that there can be several global optima for an objective function. In general, we aim to find only one of those. We can equally formalize the optimization setting as a maximization.

As a result of how we define the (sub) input and output spaces for an objective function as a task, we obtain several *task types*. We outline the four major task types for which we provide tasks and benchmark subselection in `carps`:

**black-box (BB)** : Classic task type where we can only observe the output for an input, with no notion of fidelity and optimization, and one objective.

**multi-objective (MO)** : BB task but with multiple objectives.

**multi-fidelity (MF)** : Task type, where we can query the objective function at different fidelities.

**multi-fidelity-objective (MOMF)** : Combination of MO and MF: We can query the objective function at different fidelities and aim to optimize over more than one objective.

Finally, the term *task set*, also commonly referred to as *benchmark*, is a collection of tasks. In `carps`, we provide task sets from the literature and also a task set subselection for each of the aforementioned task types.

## 2.2 Optimization Runs and Optimizers

The key idea of optimization is to evaluate the objective function to find an optimum. Approaches to this are diverse, from model-free to model-based, from parallel to sequential approaches. Before diving into a quick overview of those approaches for different task types, we define the terms history, trajectory, and aggregation. *History* denotes the sequence of evaluated trials together with the observed objective function value. *Trajectory* denotes the sequence of incumbents with the objective function value. Later, to inspect and interpret results, we use *aggregation* where we aggregate the optimizer histories or trajectories of a task set to a scalar. For example, this could mean averaging the best objective function value per optimization run over tasks. The aggregation can be performed on raw objective function values or on rankings between optimizers.

The simplest optimization algorithm is random search[2], which is a model-free approach. It can be easily parallelized and readily applied to all task types mentioned.

Then there is another important, and more sample-efficient, paradigm for optimization: Model-based and population-based approaches. In general, model-based approaches follow a sequential optimization scheme, also known as *ask and tell* or *observe and suggest* (Turner et al., 2021). First, the objective function is queried to obtain an observation. With this observation, together with all previous observations, a model is trained to approximate the task. This step is also called *tell* or *observe*. Based on this model, the next hyperparameter configuration to be evaluated is proposed. This step is also called *ask* or *suggest*. Those steps are repeated until the optimization budget, whether it be wallclock time or number of trials, is depleted. This distinction has the advantage of moving control of the optimization run one layer up, away from the optimization method, to ensure consistent benchmarking. A famous representative of a model-based approach is Bayesian optimization (Mockus, 1989).

Population-based approaches can also be formulated as sequential optimization, where after each observation, a population of hyperparameter configurations is evolved. Evolutionary algorithms like Genetic Algorithm are an example of this paradigm.

## 3 Related Work

Due to the critical role of HPO in enhancing the performance of machine learning models, several benchmarking frameworks have been developed to evaluate and compare HPO algorithms. In this section, we review existing benchmarking frameworks for HPO and previous approaches to select representative benchmarking subsets.

---

[2]Manual tuning and grid search feel also very simple and natural, but are advised against (Bergstra and Bengio, 2012).

**Benchmarking Frameworks**  We assess open-source benchmarking frameworks based on the use case, task, and optimizer diversity, and extensibility. HPOBench (Eggensperger et al., 2021) and YAHPO (Pfisterer et al., 2022) offer collections of benchmarking tasks, with a large variety of ML algorithms and datasets, but provide no integrated optimizers to compare against directly. Bayesmark (Turner and Eriksson, 2019) is designed for Bayesian optimization methods on real machine learning tasks, supporting standard machine learning algorithms on toy datasets as well as custom data, but lacks surrogate or tabular benchmarks and support for multi-objective (MO) and multi-fidelity (MF) optimization. In addition, it cannot be extended with new benchmarking tasks. HPO-B (Pineda Arango et al., 2021) focuses on benchmarking black-box HPO algorithms, featuring diverse configuration spaces on numerous datasets with both tabular benchmarks and surrogates, though it has limited optimizer support and no MO or MF capabilities. However, their tabular benchmarks do not include a search space, but only a list of available points. Additionally, their surrogate benchmarks represent categorical hyperparameters as one-hot-encoded real-valued hyperparameters, which alters the optimization task by allowing the optimizers to query points that do not exist in the original configuration space. Kurobako[3] is a command-line tool for evaluating black-box optimization algorithms. While Kurobako facilitates benchmarking across different task types, its primary integration is with Optuna (Akiba et al., 2019) and offers only a limited selection of optimizers. Furthermore, the project has seen no active development since its last release in 2022, which may limit its suitability for ongoing and future research. Synetune (Salinas et al., 2022) provides state-of-the-art algorithms with out-of-the-box tabular and surrogate benchmarks. It supports BB and MF optimization. `carps` furthermore is the only framework that offers a sub-selection of representative benchmarking tasks based ready for developing methods and reporting results. Please see Table 2 in the appendix for a compact overview.

**Benchmark Subselection**  As a universal truth, the choice of benchmark tasks significantly influences the statistical analysis of the performance. Evaluating the same set of algorithms on different sets of benchmarks might yield varying outcomes (Cenikj et al., 2022), motivating well-justified benchmark tasks. Pfisterer et al. (2022) propose subselections for the single-objective and multi-objective cases for the surrogate-based benchmark collection YAHPO-Gym. They have been selected based on the surrogate's approximation quality and task diversity; however, we aim to select a diverse and representative subset of tasks. Therefore, we aim at a principled way to subselect benchmark tasks w.r.t. diversity. This general problem of subselecting instances to compare algorithms on has been addressed in other domains. For BB tasks, many methods use meta-features based on the benchmark task, excluding performance data and subselect using unsupervised learning or graph algorithms (Eftimov et al., 2022; Cenikj et al., 2022; Ispirova et al., 2024; Dietrich et al., 2024). However, similar task features may result in vastly different algorithm performances (Nikolikj et al., 2023; Long et al., 2023). Instead of using landscape features, Cenikj et al. (2022) build a similarity graph based on trajectory features. Based on this graph, a graph algorithm selects diverse, representative, and non-redundant tasks. Whilst subselecting a diverse set of tasks, their method relies on hyperparameters of their selection method, which are not intuitive to set, and meta-features must be derived anew for each domain.

In the field of ML, there are multiple works that create subsets of benchmarks due to an abundance of potential tasks. The creators of OpenML proposed to filter datasets based on inclusion and exclusion criteria and created the OpenML-CC18 (Bischl et al., 2021). Due to the unsupervised nature of the process, there is no guarantee that the instances are complementary and cover the whole instance space. Aiming to reduce the 72 datasets of OpenML-CC18 to a minimal subset, Cardoso et al. (2021) apply item-response theory (Martínez-Plumed et al., 2020) and found that only 10 of the datasets were really hard, but also, that a subset consisting of 50% of the datasets is sufficient. The TabZilla benchmark suite (McElfresh et al., 2023) is also based on OpenML data, but uses three heuristics to include only datasets that cannot be solved by baseline algorithms, are hard for most algorithms, and are hard for gradient-boosted decision trees, thereby taking information about the current landscape of ML algorithms into account. Lastly, the Lichtenberg-MATILDA approach (Pereira et al., 2024) aims to find a maximum-coverage set of instances in a 2d projection of the instance space. This, however, requires instance or task features again. Instead, `carps` performs subselection of tasks in the performance space.

---

[3]Kurobako, https://github.com/optuna/kurobako, 2019.

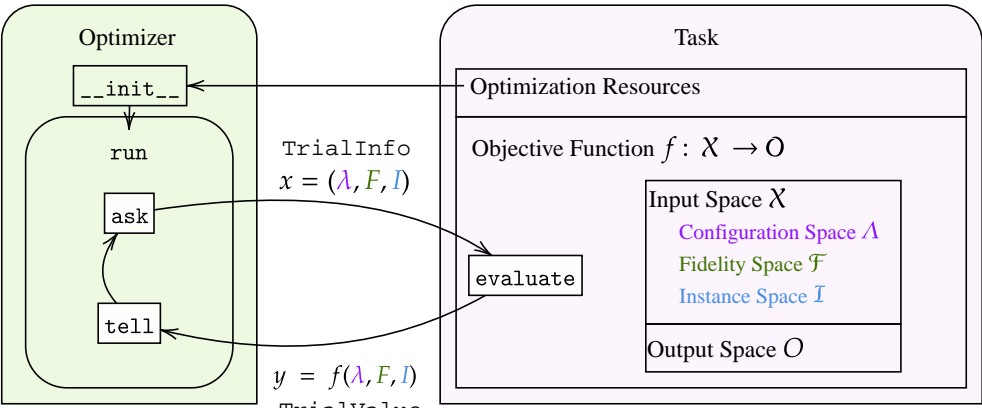

Figure 2: Overview of basic interface methods and interactions in `carps`. The `Optimizer` class orchestrates optimization and runs the optimization loop via ask-and-tell. The actual optimizer variant only needs to implement the functions `ask` and `tell`. The `Task` defines the optimization resources and holds the `ObjectiveFunction`, which is queried in between (`evaluate`). Both are configured via files using Hydra.

## 4  `carps`: Framework Overview

We designed the `carps` framework with the following desiderata in mind: First, adding your own optimizer, benchmarking, and performing evaluations is easy by providing standardized and straightforward interfaces, lowering the entrance barrier to the field. Second, representative benchmark collections for the development and testing of optimizers on each task type are offered to allow for fast prototyping and easy comparison between optimizers while requiring fewer computational resources. Third, `carps` contains an analysis pipeline to facilitate the interpretation and presentation of results. Fulfilling these desiderata ensures that `carps` is a framework that brings novel value to the community. `carps` contains 3 336 tasks from 5 benchmark collections and 26 optimizers from 9 optimizer suites.

For a more technical, in-depth view, we provide tutorials and a template repository on how to use the framework and how to add your own benchmark and optimizer in the documentation.[4]

**Reproducibility and Large-Scale Experiments**  `carps` is implemented and available under an OSS license. Experiment runs can be easily parallelized and launched via Hydra supporting SLURM (submitit), Ray, RQ, and Joblib. In addition, `carps` aims to be as reproducible as possible. For this, apart from package specification, singularity containers can be used in combination with experimentation scheduling and logging to a MySQL database as an experimental feature.

### 4.1  Interface

The interface between optimizers and tasks is kept as lean as possible while still allowing flexibility for different use cases. This is achieved by providing abstract implementations for both the optimizer and the objective function that can be subclassed. Information on evaluations is exchanged via structures holding all necessary information to perform an evaluation (`TrialInfo`) and information obtained after performing the evaluation (`TrialValue`). The classes and structure follow the conventions introduced in Section 2. An overview of the basic interface is given in Figure 2. In general, each component is specified and configured via configuration files. See Appendix F for a more technical interface description.

**Optimizer**  The optimizer orchestrates the optimization. It receives the task and, as such, must be capable of converting the configuration space (the established `ConfigSpace` (Lindauer et al., 2019)) to its own

---

[4]https://AutoML.github.io/CARP-S/latest

configuration space – in practice, this requires only minimal efforts, e.g., matching a float hyperparameter of `ConfigSpace` to the float hyperparameter of the optimizer's configuration space. The basis for the optimizer interface is then the ask-and-tell interface, wherein the `ask` method prompts the optimizer to return a new trial to evaluate (here, `TrialInfo`) and the `tell` method allows reporting back evaluation results (here, `TrialValue`). In addition, there must be a function to obtain the current incumbent, which is especially important for MF and MO optimization, where the strategy to determine an incumbent is optimizer-dependent. In the case of MO, the incumbent would be the Pareto front of configurations. Although not recommended, `carps` also allows optimizers not to implement an ask-and-tell interface; however, then comparable resource limitations cannot be guaranteed since the optimizer has to take care of it itself, which can lead to unexpected benchmarking results (Eggensperger et al., 2019).

**Objective Function and Task**   The objective function interface's mandatory methods include only two functionalities: (i) converting the objective function's configuration space into the unified configuration space for the optimizer, and (ii) evaluating a `TrialInfo` and returning results as a `TrialValue`. Optionally, if known, the global function minimum can be returned. Together with the optimization resources, the objective functions form a *task*. The optimization budget can be the number of trials (function evaluations) or time; we focus on the former. The number of trials depends on the dimensionality $d$ of the objective function and is calculated with $n_{trials} = \lceil 20 + 40\sqrt{d} \rceil$, the same as in YAHPO-Gym (Pfisterer et al., 2022).

## 5  Optimizer Overview

To facilitate easy comparisons among multiple optimizers, `carps` provides access to a wide range of optimization algorithms. Aligned with the integrated task sets, `carps` supports optimizers for the task types BB, MO, MF, and MOMF. Besides RandomSearch (Bergstra and Bengio, 2012), HEBO (Cowen-Rivers et al., 2022), Skopt[5] and DEHB (Awad et al., 2021), Ax (Olson et al., 2025) based on BoTorch (Balandat et al., 2020), multiple variants of Optuna (Akiba et al., 2019), Nevergrad (Rapin and Teytaud, 2018), SMAC3 (Lindauer et al., 2022) and Synetune (Salinas et al., 2022) are included. We use the default settings of the optimizer variants for the different task types. Table 3 in the appendix provides an overview of the optimizers in `carps` with their respective task types.

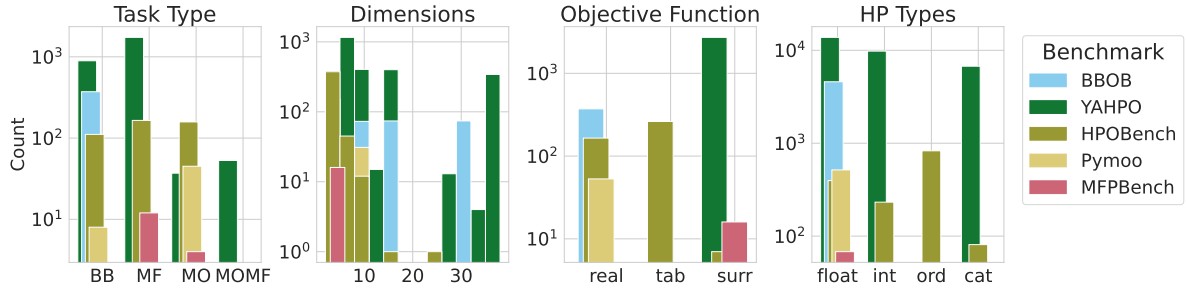

Figure 3: Statistics for all tasks included in `carps`, distinguished by benchmark collections. All benchmark collections exhibit different profiles along task types (first), dimensionality (second), objective function type (third), and hyperparameter types (fourth). As task types, we have black-box (BB), multi-fidelity (MF), multi-objective (MO), and multi-fidelity-objective (MOMF). An objective function can be a real function evaluation (real), a look-up table (tab), or a surrogate (surr). As hyperparameter (HP) types, we have continuous (float), integers (int), ordinals (ord, ordered and discrete elements), and categoricals (cat, unordered elements).

## 6  Included Task Families

With `carps`, we aim to ease accessibility to HPO tasks. We focus on four task types, namely *blackbox* (BB), *multi-fidelity* (MF), *multi-objective* (MO), and *multi-fidelity-multi-objective* (MOMF). The integrated benchmark collections offer tasks for each task type.

---

[5]Scikit Optimize, https://github.com/scikit-optimize/scikit-optimize, 2018.

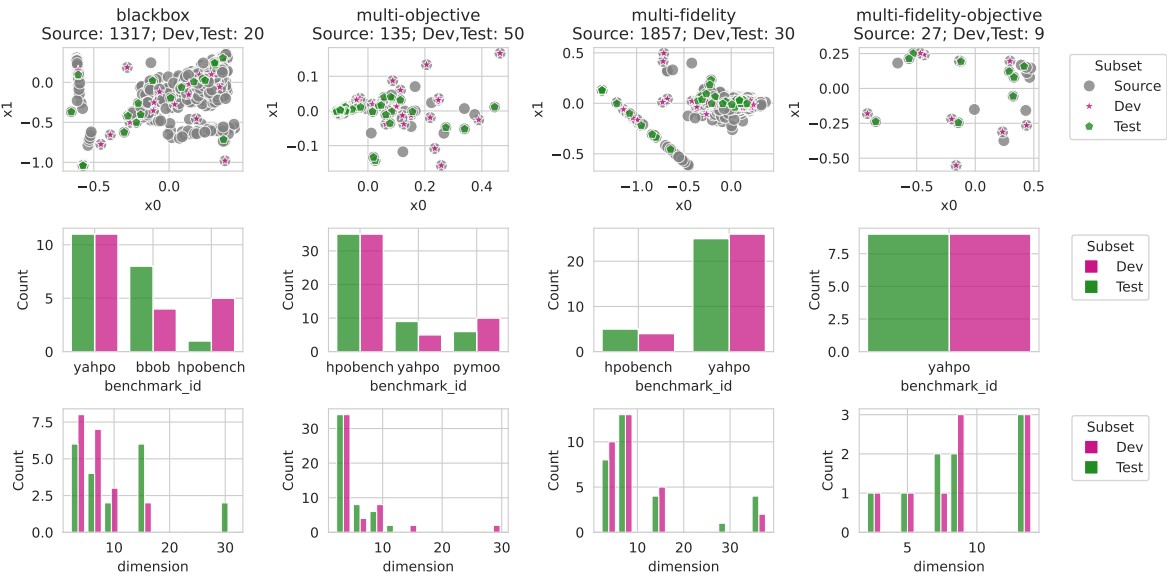

Figure 4: Subselection results summary. **Top row:** Each dot means one task and is represented by the 3-dimensional optimizer performance vector $\mathbf{y}_i$ (Equation (4)), which is reduced via PCA to 2 dimensions. The subselected tasks follow the source distribution. **Middle row:** The histogram of the selected benchmark families. **Bottom row:** Histogram of the different dimensionalities of the selected tasks.

There is a diverse set of benchmark tasks included in `carps`. Upon release, the benchmark collections provided are BBOB (Hansen et al., 2020)[6], HPOBench (Eggensperger et al., 2021), YAHPO (Pfisterer et al., 2022), MFPBench (Mallik et al., 2023) and Pymoo-MO (Blank and Deb, 2020). In addition to the task types they address, the benchmarks can be characterized by the number of dimensions, i.e., the number of hyperparameters and their types. An overview of all tasks and their respective characterization, i.e., the sum of task types, objective functions, and hyperparameter types over all tasks in the benchmark, as well as the task's dimensions, can be found in Figure 3.

## 7 Benchmark Subselection

As `carps` includes many benchmark families, with a total number of around 3 336 tasks, developing, evaluating, and reporting the performance of an optimizer can be easily biased as the distribution of tasks per benchmark collection is not equal. Furthermore, extensive evaluations on many tasks become computationally impractical due to optimizer and objective function evaluation overhead. Therefore, we propose to *subselect* representative benchmarking tasks for each task type. In addition, we propose to use two disjoint sets of benchmarking tasks, one for the development phase and one for reporting unbiased performance, similar to the commonly applied train/test split for assessing supervised ML. The next paragraphs describe the subselection methodology, the setup, and the subselection results.

### 7.1 Subselecting Representative Sets

Because of the sheer mass of benchmarking tasks, which is also not equally distributed across benchmarks (hence with the ability to strongly bias reporting of performance (Cenikj et al., 2022)), we suggest an initial subselection of representative benchmarking tasks for each task type. In addition, we propose to select two sets: one for the development of an optimization method and one for testing its performance. We can abstract the subselection as an optimization problem where, from a point cloud, we want to select those $k$ points that

---

[6]We include synthetic functions, such as BBOB and Pymoo-MO, to study performance on an established set of tasks with known characteristics. However, since we focus here on HPO tasks, we emphasize that results on these functions do not necessarily generalize to performance on actual HPO tasks and might require different search behavior (Benjamins et al., 2023).

best cover the space spanned by the point cloud. There are two avenues to create points to subselect from: (i) task features and (ii) solely from performance data. For the first option, recent work strongly suggests that state-of-the-art features for HPO and BB tasks do not capture the objective function structure well enough for AutoML approaches (Nikolikj et al., 2023; Long et al., 2023; Vermetten et al., 2023). Since this weakness of black-box optimization features has been observed by the community for several years now, without proper remedies in place, we resort to the second option, subselecting in the performance space, which reflects the behavior and performance of the optimizers. Formally, we have the set of $M$ benchmarking tasks together with the performance of $N$ optimizers:

$$P = \{\mathbf{y}_i\}_{i=1,\ldots,M}, \, \mathbf{y}_i \in [0,1]^N . \tag{4}$$

$\mathbf{y}_i$, therefore, holds the final performance of each optimizer, i.e., the performance of the incumbent, on the benchmarking task $i$. The performances are scaled to the unit-interval per task to accommodate different output scales of tasks. We want to select two subsets of $P$, each of which is as representative as possible of the initial set of $M$ tasks. Discrepancy measures are frequently used to quantify how uniformly distributed a given set of points is. Let $q$ be a $d$-dimensional vector in the unit cube $q = (q_1, \ldots, q_d) \in [0,1]^d$ and $[0,q)$ the $d$-dimensional box with a corner in the origin $[0,q_1) \times [0,q_2) \times \ldots \times [0,q_d)$. The $L_\infty$ star discrepancy of a set $P$, $d_\infty^*(P)$, measures the worst absolute difference over all such boxes $[0,q) \subseteq [0,1)$ between the Lebesgue measure of this box and the proportion of points $|P \cap [0,q)|/|P|$ that falls inside this box, see Figure 8 in the appendix for a visualization.

More formally, for a point set $P$ in dimension $d$, it is given by

$$d_\infty^*(P) := \sup_{q \in [0,1]^d} \left| \frac{|P \cap [0,q)|}{|P|} - \lambda([0,q)) \right|, \tag{5}$$

where $\lambda$ is the Lebesgue measure, $\lambda([0,q)) = \prod_{i=1}^d q_i$. For $d = 1$ it measures the length, for $d = 2$ the area, and for $d = 3$ the volume of the box. In our case, the dimension equals the number of optimizers, $d = N$. It has been used in a very wide variety of applications from computer vision to financial mathematics and Quasi-Monte Carlo integration (Paulin et al., 2022; Galanti and Jung, 1997; Santner et al., 2003; Dick and Pillichshammer, 2010). In machine learning, low discrepancy constructions such as those suggested by Sobol (Sobol, 1967) and Halton (Halton, 1964) are used for hyperparameter optimization (Bousquet et al., 2017; Cauwet et al., 2020) and to initialize surrogate-based optimization algorithms (Jones et al., 1998; Snoek et al., 2012). To obtain low discrepancy subsets, we make use of an approach proposed in (Clément et al., 2024). Interestingly, their work was originally motivated by a similar problem as ours, the search for diverse instances of the traveling salesperson problem (Neumann et al., 2018).

In our setting, the $L_\infty$ star discrepancy can be used to characterize the quality of the selected sets. Thus, one way to formulate our problem is: Find the set of $k$ points that minimizes the star discrepancy to the complete set. This corresponds to the approach to the Star Discrepancy Subset Selection Problem (Clément et al., 2022; 2024). For a description of the algorithm, please see Appendix G.

In order to determine the subset size $k$ and because optimization of the subset is resource-intensive, we optimize for $k \in \{10, 20, 30, \ldots, 100\}$ (or maximum half of the full set size) and select the $k$ points, for which the sum of the star discrepancies of the development and test subset is the lowest. To obtain two sets, the development and test set, we perform this subset selection twice: once on the $m$ initial tasks and a second time on the $m - k$ remaining ones. With the huge number of available tasks $m$, this is possible because $m - k$ is of the same order as $m$, and the distribution of the remaining $m - k$ points will still resemble that of the full set. By construction, the obtained sets represent the original set in the performance space.

For each task type, we select three diverse optimizers commonly used for the task type and record their mean performance of 20 seeds per task, forming a 3-dimensional optimizer performance vector per task $\mathbf{y}_i \in \mathbb{R}^3$ (Equation (4)), see Figure 4 top row. To perform the subselection, the performance must be normalized to the unit interval to become the source space for the subselection routine. In addition, the source space can be further log-transformed. In total, we must set two hyperparameters for the subselection routine: The subset sizes and whether to perform the subselection in the log-transformed source space or not. We select the subset

size and transformation post-hoc via filtering and decision rule: 1. Calculate ranks based on non-parametric test (Section 8.1); 2. keep combinations, where the rank stays the same between the dev and test subsets; 3. from those, keep combinations exhibiting significant differences; and 4. pick the combination filling the source space best, indicated by the sum of discrepancies of the dev and test subset.

## 7.2 Subselection Setup and Results

For the blackbox task type, we generate the source set from the performance of random search, CMA-ES, and Bayesian optimization , which are different algorithm classes commonly used for this task type. In general, one could further extend the set of optimizers to other algorithm classes, like local search algorithms, or approximative gradient optimizers, as soon as they become widely used. This introduces a bias that could hamper the ability of new algorithm classes to clearly demonstrate their performance gains. According to the aforementioned workflow, we obtain a subset of size $k = 20$ originating from the log-transformed source space. For multi-fidelity, we obtain a subset size of $k = 30$ from the log-transformed performance source space from Hyperband, DEHB, and BOHB. For multi-objective, the subset size is $k = 50$ from the non-transformed performance source space generated from running random search, Optuna-MO, and differential evolution from Nevergrad. Last but not least, for multi-fidelity-objective, the subset size is $k = 9$ from the log-transformed space of the optimizers random search, SMAC3-MOMF-GP and differential evolution from Nevergrad (the latter ignores the multi-fidelity). The selected subsets per task type are visualized together with the histogram of the task benchmark families and dimensions in Figure 4. For the complete list of tasks per task type, see Appendix H.3.

We validate the consistency of our subselections by comparing the ranks across the dev set and the test set. The ranks are determined as detailed in the analysis pipeline in the following Section 8.1. The ranking maintains its order for the subselections. In addition, we validate the representativeness of the subselection using a held-out optimizer evaluated on the full set of blackbox tasks and confirm that the ranking on the full set and on the subsets remains unchanged. See Appendix H.2 for more details.

## 8 Benchmarking with `carps`

For each task type, we put `carps` into action and describe the experimental results. We select one representative optimization method from each optimizer family and run it on the subselected task set for each task type. We run everything for 20 random seeds. Please find the code and the raw experimental results in our GitHub repository (https://www.github.com/automl/CARP-S). Running the main experiments for one optimizer requires approximately 3035 of CPU hours, with only 21 hours run for the black-box subsets; more details in Appendix E.

## 8.1 Analysis Pipeline

We analyze experimental results from different viewpoints. Most importantly, we aggregate the results via rankings. We use the library `autorank` (Herbold, 2020) for determining the ranks and critical differences. The ranking is performed on the raw performance values, averaged across seeds. To calculate the performance metric in the case of MO, we first calculate the reference point per task across all collected runs as the maximum of each objective. Then, for each run, we calculate the hypervolume iteratively using all tuples of observed objective values. To be more precise regarding the ranking, we use the frequentist approach (Demšar, 2006): We use the non-parametric Friedman test as an omnibus test to determine whether there are any significant differences between the median values of the populations. We use this test because we have more than two populations, which cannot be assumed to be normally distributed. We use the post hoc Nemenyi test to infer which differences are significant. The significance level is $\alpha = 0.05$. In order to be considered different, the difference between the mean ranks of two optimizers must be greater than the critical difference (CD). We visualize the ranks for the final incumbent with indication of CD (Figure 5a) and the ranks over time (Figure 5b). In addition, we also show the performance per task and optimizer as a heatmap (Figure 6). Please find results for all task types and subsets in Appendix I.

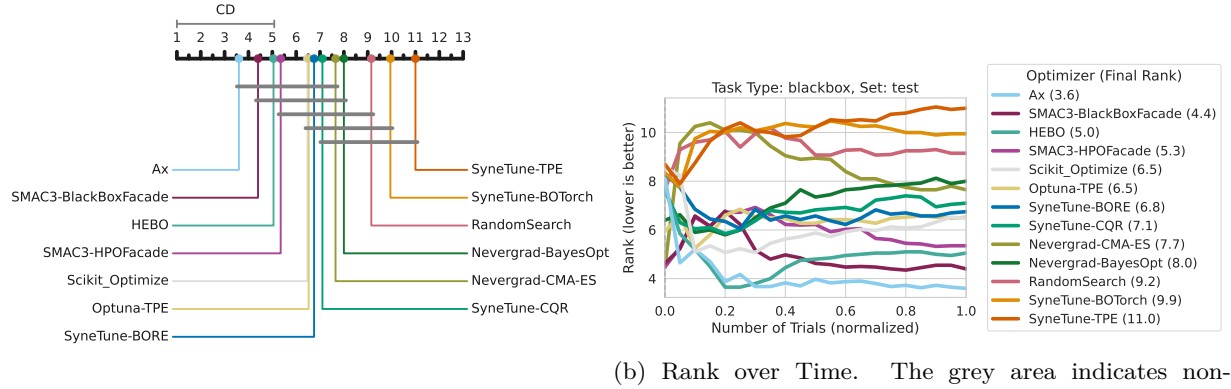

(a) Critical Difference Diagram

(b) Rank over Time. The grey area indicates non-significance.

Figure 5: Blackbox task type (test set)

## 8.2 Insights

For each task type, many optimizers are not statistically significantly different. Their performance does differ on different tasks, indicating a possible complementarity. For example, in the black-box test set on BBOB task (function id 11, dimension 32), Nevergrad-CMA-ES excels whilst otherwise performing mediocre (see Figure 6). Critical difference plots for all task type (subset test) are found in Figure 5a and Section 10. Sometimes, for the black-box task type, the order of ranks differs across the dev and test sets. However, the distance of the ranks in those cases is lower than the critical difference; thus, the absolute rank and the order thereof should be taken with a grain of salt. Upon inspection of the composition of the task type subsets, there is a mixture of easy and harder task instances, and optimizers perform in general similarly on them. On the topic of anytime performance, the rankings mostly stabilize after 60%-80% of the optimization budget, sometimes with great differences to the final rank in the first few trials.

Our recommendation for research on optimization algorithms is the following: (i) define a task area where to advance (for example, very high-dimensional black-box optimization or specific application domains), (ii) develop in this task area, (iii) verify general performance with a proposed subset, (iv) consider different ways to inspect results, and (v) carefully report for the task area and the general case without generalizing conclusions.

## 9 Limitations and Future Work

So far, our benchmarking framework `carps` focuses on benchmarking HPO. In principle, complete AutoML for pipeline construction could be benchmarked as well (Olson and Moore, 2019; Feurer et al., 2022). A limitation is that `carps` depends on configuration spaces that can be expressed via `ConfigSpace` (Lindauer et al., 2019) and thus is not compatible with unbounded configuration spaces defined by grammars or generic pipelines. Motivated by their theoretically proven advantages in Monte Carlo methods, we have used in this work the $L_\infty$ star discrepancy as a criterion for the selection of representative instances. Many other diversity metrics exist and could possibly be considered. For example, a property of the $L_\infty$ star discrepancy that we currently do not know how to fairly assess is its tendency to select more points in the upper right corner. Recent work has shown that this bias can be avoided at almost no cost by considering symmetrized versions of the $L_\infty$ star discrepancy notion (Clément et al., 2025). To compare such alternative diversity measures for our benchmarking purposes, we would need to have efficient subset selection methods, which are currently not available.

In the future, we plan to extend our framework to include more optimizers and benchmarks steadily. Furthermore, we would like to extend our task types to include constraints and plan to integrate a parallel execution system, allowing fair benchmarking, which is non-trivial, as proposed by Watanabe et al. (2024). This is relevant as some optimizers, like ASHA (Li et al., 2020), only then unleash their full potential. In

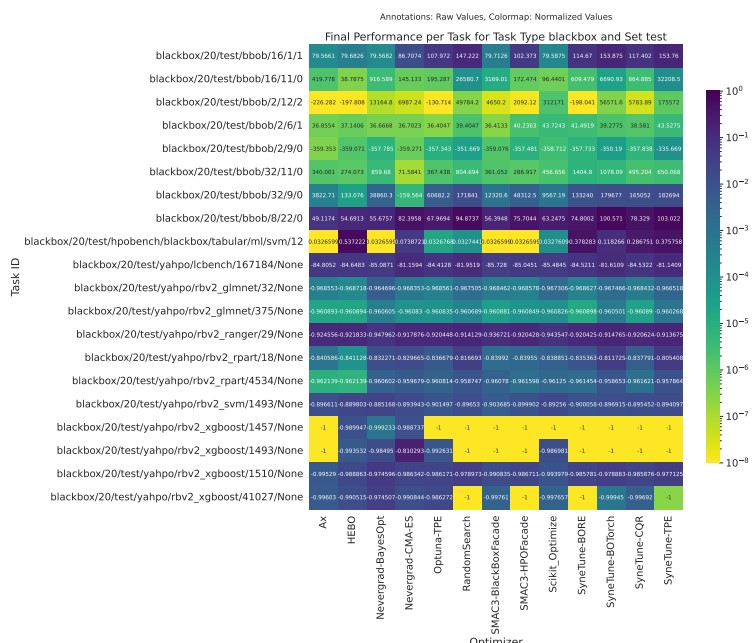

Figure 6: Blackbox task type (test set): Final incumbent cost per task, averaged over 20 seeds.

addition, `carps`, a framework holding different benchmark collections, can be a stepping stone to active benchmarking, i.e., instead of evaluating an algorithm on all available tasks, the tasks are selected actively in order to build a holistic and nuanced view of the algorithm's performance, also regarding different task types. Our work currently uses algorithm performances to span a task space, and we plan to contrast that with also using meta-features describing the tasks, possibly combining both (Sim and Hart, 2022). This would enable characterizing the strengths and weaknesses of optimizers on specific task types.

## 10   Conclusion

With `carps`, we propose a lightweight benchmarking framework for HPO. It offers numerous benchmarks and optimizers as baselines and is conceptualized to facilitate extension and scalability. We target four task types, namely blackbox, multi-fidelity, multi-objective, and multi-fidelity-multi-objective. For those task types, we propose an initial subselection of representative benchmarking tasks, one for development and one to test an optimizer's performance, along with the inclusion of functionality to recompute these subsets as more benchmarks become available. Together with an analysis pipeline, `carps` offers everything needed to develop new HPO optimizers.

### Acknowledgements

Sarah Segel, Helena Graf, and Marius Lindauer gratefully acknowledge funding by the European Union (ERC, "ixAutoML", grant no. 101041029). The authors gratefully acknowledge the computing time provided to them on the high-performance computers Noctua2 at the NHR Center PC2 under the project hpc-prf-intexml. These are funded by the Federal Ministry of Education and Research and the state governments participating on the basis of the resolutions of the GWK for the national high-performance computing at universities (www.nhr-verein.de/unsere-partner). Carolin Benjamins, Carola Doerr, and Marius Lindauer acknowledge funding by the French National Research Agency (ANR-23-CE23-0035) and the German Research Foundation (DFG; LI 2801/7-1), through project OPT4DAC. Carola Doerr and Deyao Chen acknowledge funding by the CNRS IEA DynAC project and by the European Union (ERC, "dynaBBO", grant no. 101125586). This work was granted access to the HPC resources of the SACADO MeSU platform at Sorbonne Université. Katharina Eggensperger acknowledges funding by the Deutsche Forschungsgemeinschaft

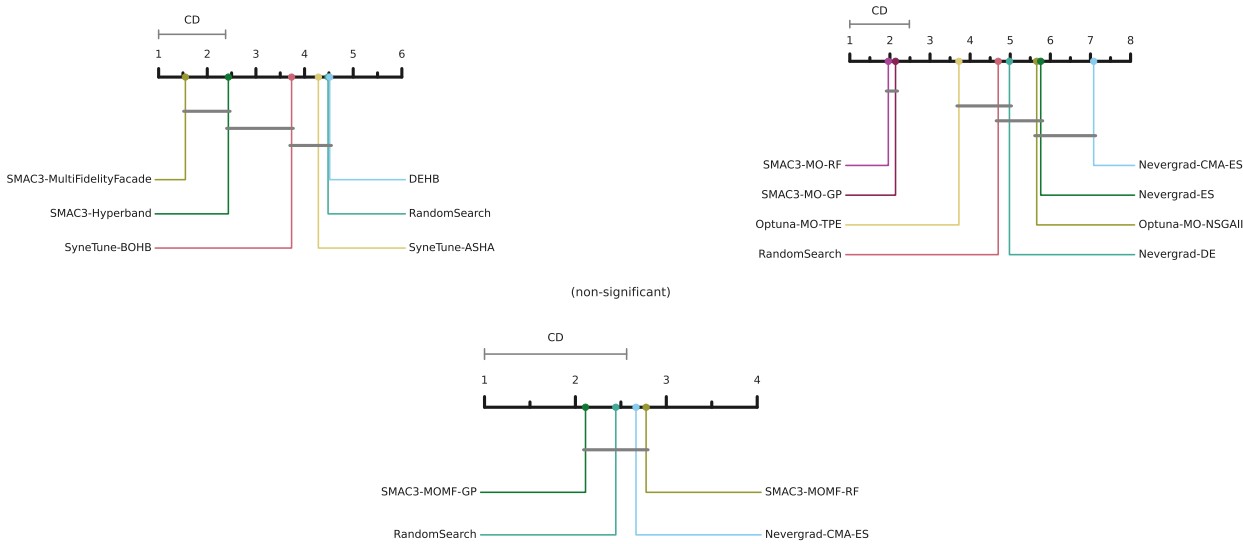

Figure 7: Critical Difference Diagrams. **Upper left:** Multi-fidelity task type (test set). **Upper right:** Multi-objective task type (test set). **Bottom:** Multi-fidelity-objective task type (test set).

(DFG, German Research Foundation) under Germany's Excellence Strategy – EXC number 2064/1 – Project number 390727645. Edward Bergman and Frank Hutter acknowledge TAILOR, a project funded by the EU Horizon 2020 research and innovation program under GA No 952215. Neeratyoy Mallik and Frank Hutter also gratefully acknowledge funding by the European Union (ERC, "Deep Learning 2.0", grant no. 101045765). Views and opinions expressed are, however, those of the author(s) only and do not necessarily reflect those of the European Union or the European Research Council Executive Agency. Neither the European Union nor the granting authority can be held responsible for them.

### Broader Impact

We do not foresee direct negative societal impact from our work since we do not target specific applications. Overall, AutoML and HPO aim to democratize ML and AI further, which comes with all the general benefits and risks of making AI available to everyone. In general, `carps` eases the research and computational burden of developing optimizers for HPO, and eventually contributes to better scientific practices.

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

## 11 Glossary

| Term | Definition |
| --- | --- |
| HPO | Paradigm: Hyperparameter Optimization (Feurer and Hutter, 2019; Bischl et al., 2023) |
| ML | Machine Learning |
| BB | Paradigm: Blackbox; an optimization task type in which solely inputs and outputs of the optimization object are available. |
| MF | Paradigm: Multi-Fidelity; an optimization task type with cheaper approximations of the objective function available. |
| MO | Paradigm: Multi-Objective; an optimization task type, where more then one objective need to be optimized. |
| MOMF | Paradigm: Multi-Fidelity Multi-Objective optimization. |
| HPOBench | Benchmark: Hyperparameter Optimization benchmark (Eggensperger et al., 2021) |
| YAHPO-Gym | Benchmark: Yet Another Hyperparameter Optimization Gym (Pfisterer et al., 2022) |
| HPO-B | Benchmark: A Large-Scale Reproducible Benchmark for Black-Box HPO based on OpenML. (Pineda Arango et al., 2021) |
| MFPBench | Benchmark: A multi-fidleity prior benchmark. (Mallik et al., 2023) |
| BBOB | Benchmark: Black-box optimization benchmarking. (Hansen et al., 2020) |
| OpenML-CC18 | Benchmark: OpenML curated classification tasks. (Bischl et al., 2021) |
| Pymoo-MO | Benchmark: Multi-objective Optimization in Python (Blank and Deb, 2020) |
| CARPS | Benchmark: Comprehensive Automated Research Performance Studies (ours) |
| ASHA | Optimizer: Asynchronous Successive Halving (Li et al., 2020) |
| DEHB | Optimizer: Differential Evolution Hyperband (Awad et al., 2021) |
| BOHB | Optimizer: Bayesian Optimization Hyperband (Falkner et al., 2018) |
| HEBO | Optimizer: Heteroscedastic and Evolutionary Bayesian Optimisation solver (Cowen-Rivers et al., 2022) |
| Skopt | Optimizer: Scikit Optimize [7] |
| SMAC3 | Optimizer: Sequential Model-based Algorithm Configuration (Lindauer et al., 2022) |
| SMAC3-MOMF-GP | Optimizer: SMAC3 Multi-Objective Multi-fidelity Gaussian Process |
| Nevergrad-DE | Optimizer: Differential Evolution (Storn and Price, 1997) Strategy implemented in Nevergrad (Rapin and Teytaud, 2018) |
| Nevergrad-CMA-ES | Optimizer: CMA-ES (Hansen et al., 2003) Strategy implemented in Nevergrad (Rapin and Teytaud, 2018) |
| Optuna-MO | Optimizer: Multi-objective optimization implemented in Optuna (Akiba et al., 2019) |
| CMA-ES | Covariance Matrix Adaptation Evolution Strategy |
| CD | Critical difference. |
| Lichtenberg-MATILDA | |
| RQ | Redis Queue (RQ) |

## A Appendix

The appendix is structured as follows. We detail benchmarking frameworks in Appendix B, optimizers variants in Appendix C and the maintenance plan in Appendix D. Further, we discuss computational resources used (Appendix E) and technical details of the interface (Appendix F) and of the optimization of the star

---

[7]Scikit Optimize, https://github.com/scikit-optimize/scikit-optimize, 2018.

discrepancy (Appendix G). Details on the subselections per task type and results can be found in Appendix H and Appendix I.

## B  Benchmark Frameworks

We provide a compact overview of benchmarking frameworks in Table 2.

## C  Optimizer Overview

See Table 3 for an overview of optimizers.

## D  Maintenance Plan

Following (Eggensperger et al., 2021) and (Pfisterer et al., 2022) we describe our maintenance plan of `carps`.

**Who Maintains** `carps` is developed and maintained by the Institute of AI (Leibniz University Hannover).

**Contact** Questions and issues regarding the repository and code can be posted in the GitHub repository (`https://www.github.com/automl/CARP-S`). Other questions can be asked via the provided email.

**Erratum** There is no erratum.

**Library Updates** We plan on updating the library with new features for experiment running, analysis improvement and more optimizers and benchmarks, potentially via external pull requests. Changes will be communicated via Github releases as well as a CHANGELOG.

**Support for Older Versions** Older versions of `carps` will be available on GitHub but with limited support.

**Contributions** Contributions to our benchmarking framework `carps` are very welcome. These can either be general features or more optimizers and benchmarks. For the latter, we provide a tutorial for contributing a benchmark (`https://automl.github.io/CARP-S/latest/contributing/contributing-a-benchmark/`) with an example repository (https://github.com/automl/OptBench), and one for contributing an optimizer (`https://automl.github.io/CARP-S/latest/contributing/contributing-an-optimizer/`) with a template repository (`https://github.com/automl/CARP-S-template/blob/main/my-optimizer.py`). See `https://automl.github.io/CARP-S/latest/contributing/` for a guide how to contribute. Contributions are managed via pull requests.

**Dependencies** Python package dependencies are accessible in `pyproject.toml` in the repository. The `carps` version used for the experiments is 1.0.0.

## E  Computational Resources

All experiments were run on single CPUs of type AMD Milan 7763, 2.45 GHz, up to 3.5 GHz, each 2x 64 cores, 128GB main memory. The estimated runtime for the experiments for one optimizer on the subselection is 3035 CPU hours, see Table 4.

## F  Interface: Technical Details

To enable communication between the optimizer and the objective functions, we propose a standardized interface. This interface relies on `ConfigSpace.ConfigurationSpace` as the representation of the configuration space and `TrialInfo` and `TrialValue` to hold information about a trial. A trial info is associated with the HP configuration, the budget for multi-fidelity, the instance in the case of algorithm configuration, and an optional seed. In addition, a name can be associated with a trial info as well as a checkpoint path. The benchmark task requires a definition of the configuration space and must provide the function `evaluate`. The optimizer class holds one objective function and requires ask-and-tell to be implemented. In the case that the underlying optimizer does not support ask-and-tell, its normal run method can be used.

Table 2: Benchmarking Frameworks

| Framework | Use Case | Tasks | Optimizers | Task Types | Extensibility |
|---|---|---|---|---|---|
| Bayesmark (Turner and Eriksson, 2019) | Bayesian optimization on real ML tasks | Cross-product of 9 ML algorithms on 6 Sklearn toy datasets | 10 optimizers | BB | Add new optimizers via wrappers, datasets via CSV files, no new ML algorithms |
| HPO-B (Pineda Arango et al., 2021) | Benchmarking black-box HPO algorithms with tabular and surrogate benchmarks | 16 configuration spaces on 101 OpenML-based datasets (HPO-B-v2) | 4 optimizers | BB | Add optimizers via wrappers |
| HPOBench (Eggensperger et al., 2021) | Collection of multi-fidelity benchmarking tasks for HPO | 12 benchmark families with 110 benchmarking tasks | No included optimizers | BB, MO, MF | Add new benchmarks via adding a new benchmark class, tutorial on this |
| Kurobako[8] | Benchmarking black-box optimization | NAS-Bench-101, HPOBench, Sigopt Evalset, Two-objective ZDT functions | 4 optimizers | BB, MO, MF | Add new benchmarks and optimizers via implementation as a command line program |
| Synetune (Salinas et al., 2022) | SOTA HPO with tabulated and surrogate benchmarks | NASBench201, FCNet, LCBench, HPO-B, TabRepo | 10 optimizers | BB, MF | Add new benchmarks via code contribution and optimizers via wrappers, tutorials for both |
| YAHPO (Pfisterer et al., 2022) | Collection of benchmarking tasks for HPO and black-box optimization methods | 14 ML algorithms with 852 benchmarking tasks | No included optimizers | BB, MO, MF, MOMF | Add new benchmarks via adding benchmark configuration and meta-data, tutorial on this |
| **carps (ours)** | HPO for BB, MO, MF, and MOMF with representative benchmarking tasks | HPOBench, YAHPO, MFPBench, BBOB, Pymoo-MO. Subselections based on performance data | 26 optimizers from different frameworks | BB, MO, MF, MOMF | Add new benchmarks and optimizers via wrappers and tutorials for both |

## G  Optimization of the Star Discrepancy: Technical Details

Subset Selection was formally introduced in (Clément et al., 2022) with exact methods in dimension 2, and extended by heuristics in (Clément et al., 2024) for higher dimensions and any number of points. Since our test cases are in dimension 3 (because we describe every task with the performance of three diverse

Table 3: Optimizers included in `carps`

| Optimizer | Variant | BB | MO | MF | MO-MF |
|---|---|---|---|---|---|
| RandomSearch | | ✓ | | | |
| HEBO | | ✓ | | | |
| Skopt | | ✓ | | | |
| Ax | | ✓ | | | |
| DEHB | | | | ✓ | |
| Optuna | Optuna-TPE | ✓ | | | |
| | Optuna-MO-TPE | | ✓ | | |
| | Optuna-MO-NSGAII | | ✓ | | |
| SMAC3-2.0 | BlackBoxFacade | ✓ | | | |
| | MO (ParEGO) (Knowles, 2006) | | ✓ | | |
| | MultiFidelityFacade | | | ✓ | |
| | Hyperband (Li et al., 2018) | | | ✓ | |
| | MultiFidelityFacade (GP or RF) with MO (ParEGO) (Knowles, 2006) | | | | ✓ |
| Nevergrad | NGOpt | ✓ | | | |
| | NoisyBandit | ✓ | | | |
| | BayesOpt (Nogueira, 2014) | ✓ | | | |
| | Hyperopt (Bergstra et al., 2013) | ✓ | | | |
| | CMA-ES (Hansen et al., 2019) | ✓ | | | |
| | EvolutionStrategy | | ✓ | | |
| | DifferentialEvolution | | ✓ | | |
| SyneTune | Conformal Quantile Regression (CQR) (Salinas et al., 2023) | ✓ | | | |
| | BORE (Tiao et al., 2021) | ✓ | | | |
| | BOTorch (Balandat et al., 2020) | ✓ | | | |
| | TPE (Bergstra et al., 2011) | ✓ | | | |
| | ASHA (Li et al., 2020) | | | ✓ | |
| | BOHB (Falkner et al., 2018) | | | ✓ | |

Table 4: Runtimes in CPU Hours per Task Type

| | time |
|---|---|
| blackbox | 21 |
| multi-fidelity | 2589 |
| multi-fidelity-objective | 95 |
| multi-objective | 330 |
| total | 3035 |

optimization methods – for the black box task types it would be random search, Bayesian optimization and CMA-ES) with thousands of data points, we use the heuristics from (Clément et al., 2024). While multiple versions of the heuristic are introduced, the general principle behind each of them is the same. For the general principle see Figure 8. It is an iterative process where, at each step, one point in the current best subset is replaced by a point that is not yet selected. If the discrepancy of the new set is lower than that of the previous, the new set is kept for the next step. Since there are a very large number of possible point exchanges at each step, a careful selection of both the outgoing and the incoming points has to be made to avoid numerous expensive discrepancy calculations. The outgoing point is chosen as one of the points defining the box with the worst local discrepancy (see Niederreiter (1972) for a description of the discrete structure of the $L_\infty$ star discrepancy calculation). This point is associated with the discrepancy value for a

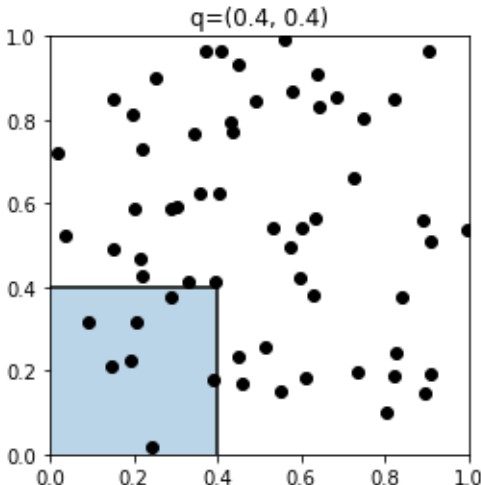

Figure 8: The discrepancy for the box defined by $q$ is given by the volume, 0.16, minus the proportion of points inside $[0, q)$, 7/60. It gives a value of 0.0433. Shifting $q$ over the entire space and keeping the worst discrepancy value gives the discrepancy value of the point set $P$.

specific dimension; the incoming point is then selected from neighboring points in this dimension. In the "nobrute" implementations, the heuristic stops once, for a given set, all such swaps have been performed and no improvement on the discrepancy could be made. For the other implementations, a further brute force verification of all possible remaining swaps is made to guarantee that the final set is a local optimum for our heuristic. The heuristic then returns the subset and its associated discrepancy value. In all cases, the result of a single run of the heuristic is highly dependent on the initial, random set used to start the heuristic. A large number of runs are therefore performed with the best (discrepancy value, associated set) combination kept. This is particularly important when $n$ is much larger than the number of selected points $k$, as is the case in our setting.

## H  Subselection

We cover additional information about the subselections per task type, i.e., show the different subset sizes, validate the ranking across subsets, visualize the subsets, and show statistics, as well as list the tasks for the dev and test subsets.

### H.1  Different Subset Sizes

For the blackbox task type, we optimize the star discrepancy for different $k \in \{20, 30, \ldots, 100\}$. We choose $k = 30$ because the discrepancy sum of both sets, dev and test, is the lowest, see Figure 9a. For the multi-fidelity task type we choose $k = 20$, see Figure 9b, and for multi-objective $k = 10$ (Figure 9c).

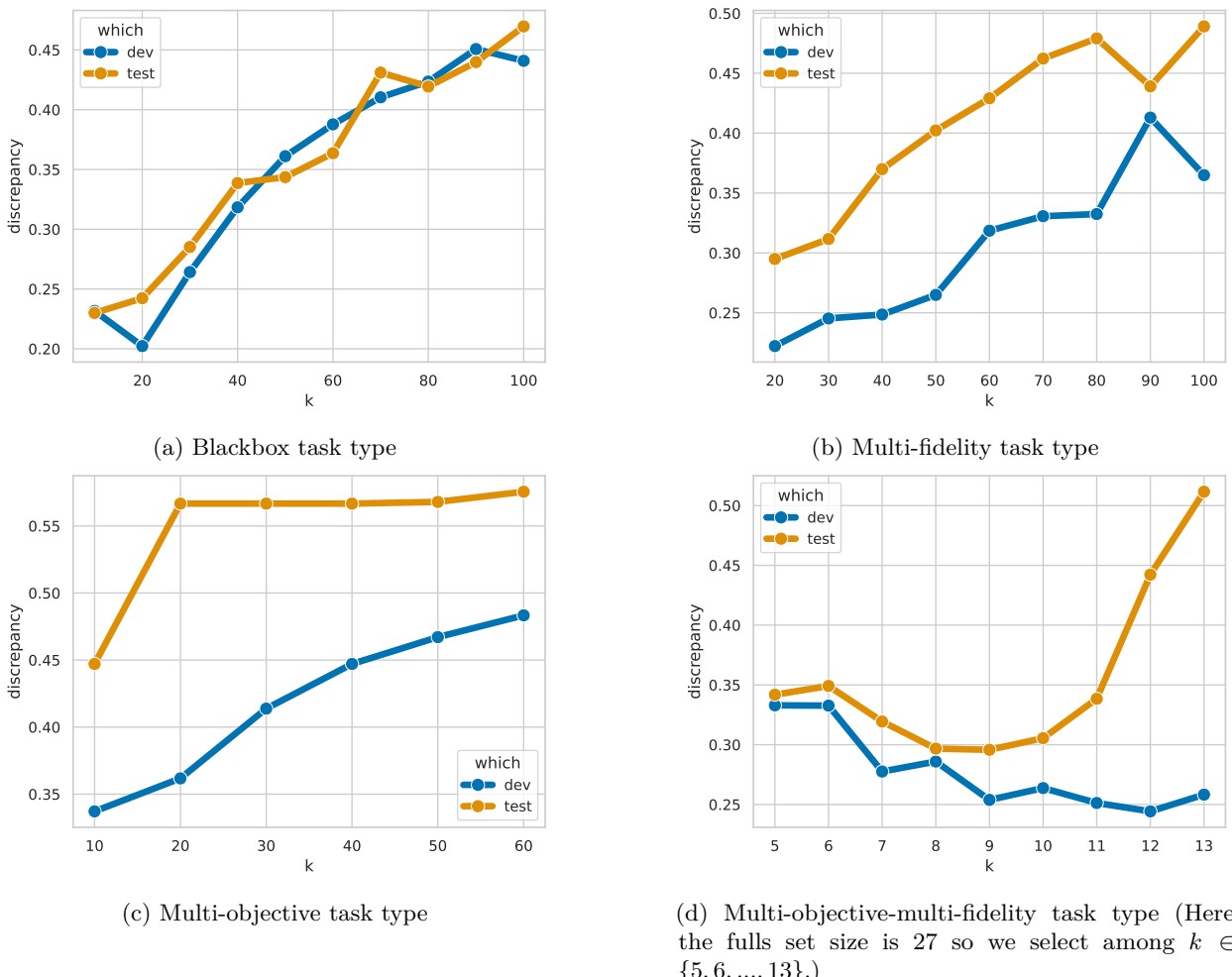

(a) Blackbox task type

(b) Multi-fidelity task type

(c) Multi-objective task type

(d) Multi-objective-multi-fidelity task type (Here the fulls set size is 27 so we select among $k \in \{5, 6, ..., 13\}$.)

Figure 9: Different subset sizes $k$ with the star discrepancy

## H.2 Ranking Validation

We validate the subselections two-fold. First, we assure consistent ranking for the dev and test sets, and second, we assure consistent ranking on the full blackbox set and on its subsets with a held-out optimizer.

In order to validate that the ranking remains consistent across the subsets, we calculate the rank the same way as described in Section 8.1 for each task type and then determine the order. The following tables (Table 5, Table 6, Table 8, Table 7) show that the ranking is consistent across the subsets for each task type.

Table 5: Mean Ranking for Scenario blackbox

| optimizer_id set_id | Nevergrad-CMA-ES | SMAC3-BlackBoxFacade | RandomSearch | significant |
|---|---|---|---|---|
| dev | 1.65 (2) | 1.65 (2) | 2.70 (3) | yes |
| test | 1.82 (2) | 1.50 (1) | 2.67 (3) | yes |

Table 6: Mean Ranking for Scenario multi-fidelity

| optimizer_id set_id | SMAC3-MultiFidelityFacade | DEHB | SMAC3-Hyperband | significant |
|---|---|---|---|---|
| dev | 1.35 (1) | 2.23 (2) | 2.42 (3) | yes |
| test | 1.57 (1) | 2.15 (2) | 2.27 (3) | yes |

Table 7: Mean Ranking for Scenario multi-fidelity-objective

| optimizer_id set_id | SMAC3-MOMF-GP | RandomSearch | Nevergrad-DE | significant |
|---|---|---|---|---|
| dev | 1.56 (1) | 1.78 (2) | 2.67 (3) | yes |
| test | 1.44 (1) | 2.00 (2) | 2.56 (3) | no |

Table 8: Mean Ranking for Scenario multi-objective

| optimizer_id set_id | Optuna-MO-TPE | Nevergrad-DE | RandomSearch | significant |
|---|---|---|---|---|
| dev | 1.30 (1) | 2.20 (2) | 2.50 (3) | yes |
| test | 1.80 (2) | 1.70 (1) | 2.50 (3) | no |

Table 9: Validating the representativeness of the subselection for scenario blackbox with a held-out optimizer (Ax). The ranking remains consistent.

| optimizer_id set_id | Ax | SMAC3-BlackBoxFacade | Nevergrad-CMA-ES | RandomSearch | significant |
|---|---|---|---|---|---|
| full | 1.84 (1) | 1.95 (2) | 2.49 (3) | 3.71 (4) | yes |
| dev | 1.64 (1) | 2.00 (2) | 2.82 (3) | 3.55 (4) | yes |
| test | 2.00 (1) | 2.00 (2) | 2.86 (3) | 3.14 (4) | yes |

To validate the representativeness of the subsets, we exemplarily evaluate a held-out optimizer on the full set of blackbox tasks. We choose Ax, which is a framework for Bayesian optimization, as is SMAC3, and thus is a member of the represented algorithm classes. We confirm that the ranking on the full set, and on the dev and test sets, remains the same, see Table 9. For the test subset, the order stays the same, although the mean rankings of the two best are almost equal. For this reason, we also consider a different view of performance. Figure 10 shows the performance in terms of normalized costs averaged across tasks and seeds for the different sets, and of the source optimizers and held-out optimizer. It also reveals that, in terms of this view of performance, the ranking remains consistent across sets. The difference here from the rank calculation is that, for ranks, the performance of each optimizer is compared *per task* with the estimated (unnormalized) performance for that task. Both views, the ranks and the normalized performance, validate that the ranking remains consistent for a held-out optimizer for the blackbox sets.

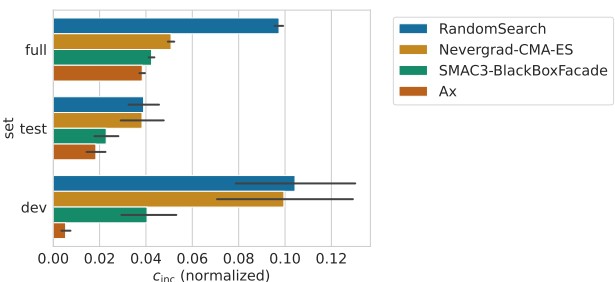

Figure 10: The normalized cost averaged over tasks and seeds of the three optimizers used for creating the blackbox subset, and one held-out optimizer (Ax). The ranking remains the same on the different sets.

### H.3  Selected Sets

Table 10: Details of Subselection

| Task Type | Statistics | Task List (dev) | Task List (test) |
|---|---|---|---|
| Blackbox | Figure 4 | Table 11 | Table 12 |
| Multi-Fidelity | Figure 4 | Table 13 | Table 14 |
| Multi-Objective | Figure 4 | Table 15 | Table 18 |
| Multi-Objective-Multi-Fidelity | Figure 4 | Table 17 | Table 18 |

Table 11: Selected tasks ('blackbox', 'dev')

| benchmark_id | task | dimensions | n_trials | n_floats | n_integers | n_categoricals | n_ordinals |
|---|---|---|---|---|---|---|---|
| YAHPO | blackbox/20/dev/yahpo/rbv2_xgboost/23512/None | 14 | 170 | 10 | 2 | 2 | 0 |
| HPOBench | blackbox/20/dev/hpobench/bb/tab/ml/lr/146818 | 2 | 77 | 0 | 0 | 0 | 2 |
| YAHPO | blackbox/20/dev/yahpo/rbv2_aknn/458/None | 6 | 118 | 0 | 4 | 2 | 0 |
| HPOBench | blackbox/20/dev/hpobench/bb/tab/nas/SliceLocalizationBenchmark | 9 | 140 | 0 | 0 | 3 | 6 |
| BBOB | blackbox/20/dev/bbob/noiseless/2/12/0 | 2 | 77 | 2 | 0 | 0 | 0 |
| BBOB | blackbox/20/dev/bbob/noiseless/2/20/0 | 2 | 77 | 2 | 0 | 0 | 0 |
| YAHPO | blackbox/20/dev/yahpo/rbv2_ranger/40927/None | 8 | 134 | 2 | 3 | 3 | 0 |
| YAHPO | blackbox/20/dev/yahpo/rbv2_svm/24/None | 6 | 118 | 3 | 1 | 2 | 0 |
| HPOBench | blackbox/20/dev/hpobench/bb/tab/nas/NavalPropulsionBenchmark | 9 | 140 | 0 | 0 | 3 | 6 |
| HPOBench | blackbox/20/dev/hpobench/bb/tab/ml/xgboost/146212 | 4 | 100 | 0 | 0 | 0 | 4 |
| YAHPO | blackbox/20/dev/yahpo/rbv2_svm/182/None | 6 | 118 | 3 | 1 | 2 | 0 |
| YAHPO | blackbox/20/dev/yahpo/rbv2_xgboost/42/None | 14 | 170 | 10 | 2 | 2 | 0 |
| YAHPO | blackbox/20/dev/yahpo/rbv2_aknn/312/None | 6 | 118 | 0 | 4 | 2 | 0 |
| YAHPO | blackbox/20/dev/yahpo/rbv2_aknn/40498/None | 6 | 118 | 0 | 4 | 2 | 0 |
| YAHPO | blackbox/20/dev/yahpo/rbv2_glmnet/41157/None | 3 | 90 | 2 | 0 | 1 | 0 |
| HPOBench | blackbox/20/dev/hpobench/bb/tab/ml/rf/146212 | 4 | 100 | 0 | 0 | 0 | 4 |
| BBOB | blackbox/20/dev/bbob/noiseless/4/6/1 | 4 | 100 | 4 | 0 | 0 | 0 |
| YAHPO | blackbox/20/dev/yahpo/rbv2_aknn/1462/None | 6 | 118 | 0 | 4 | 2 | 0 |
| YAHPO | blackbox/20/dev/yahpo/lcbench/168335/None | 7 | 126 | 4 | 3 | 0 | 0 |
| BBOB | blackbox/20/dev/bbob/noiseless/2/12/1 | 2 | 77 | 2 | 0 | 0 | 0 |

Table 12: Selected tasks ('blackbox', 'test')

| benchmark_id | task | dimensions | n_trials | n_floats | n_integers | n_categoricals | n_ordinals |
|---|---|---|---|---|---|---|---|
| YAHPO | blackbox/20/test/yahpo/lcbench/167184/None | 7 | 126 | 4 | 3 | 0 | 0 |
| BBOB | blackbox/20/test/bbob/noiseless/16/11/0 | 16 | 180 | 16 | 0 | 0 | 0 |
| BBOB | blackbox/20/test/bbob/noiseless/2/6/1 | 2 | 77 | 2 | 0 | 0 | 0 |
| BBOB | blackbox/20/test/bbob/noiseless/16/1/1 | 16 | 180 | 16 | 0 | 0 | 0 |
| YAHPO | blackbox/20/test/yahpo/rbv2_ranger/29/None | 8 | 134 | 2 | 3 | 3 | 0 |
| BBOB | blackbox/20/test/bbob/noiseless/32/9/0 | 32 | 247 | 32 | 0 | 0 | 0 |
| BBOB | blackbox/20/test/bbob/noiseless/32/11/0 | 32 | 247 | 32 | 0 | 0 | 0 |
| YAHPO | blackbox/20/test/yahpo/rbv2_svm/1493/None | 6 | 118 | 3 | 1 | 2 | 0 |
| BBOB | blackbox/20/test/bbob/noiseless/8/22/0 | 8 | 134 | 8 | 0 | 0 | 0 |
| HPOBench | blackbox/20/test/hpobench/bb/tab/ml/svm/12 | 2 | 77 | 0 | 0 | 0 | 2 |
| YAHPO | blackbox/20/test/yahpo/rbv2_xgboost/1457/None | 14 | 170 | 10 | 2 | 2 | 0 |
| YAHPO | blackbox/20/test/yahpo/rbv2_xgboost/1510/None | 14 | 170 | 10 | 2 | 2 | 0 |
| YAHPO | blackbox/20/test/yahpo/rbv2_xgboost/41027/None | 14 | 170 | 10 | 2 | 2 | 0 |
| YAHPO | blackbox/20/test/yahpo/rbv2_glmnet/32/None | 3 | 90 | 2 | 0 | 1 | 0 |
| YAHPO | blackbox/20/test/yahpo/rbv2_rpart/18/None | 5 | 110 | 1 | 3 | 1 | 0 |
| YAHPO | blackbox/20/test/yahpo/rbv2_rpart/4534/None | 5 | 110 | 1 | 3 | 1 | 0 |
| BBOB | blackbox/20/test/bbob/noiseless/2/9/0 | 2 | 77 | 2 | 0 | 0 | 0 |
| YAHPO | blackbox/20/test/yahpo/rbv2_glmnet/375/None | 3 | 90 | 2 | 0 | 1 | 0 |
| YAHPO | blackbox/20/test/yahpo/rbv2_xgboost/1493/None | 14 | 170 | 10 | 2 | 2 | 0 |
| BBOB | blackbox/20/test/bbob/noiseless/2/12/2 | 2 | 77 | 2 | 0 | 0 | 0 |

Table 13: Selected tasks ('multi-fidelity', 'dev')

| benchmark_id | task | dimensions | n_trials | n_floats | n_integers | n_categoricals | n_ordinals | fidelity_type | min_budget | max_budget |
|---|---|---|---|---|---|---|---|---|---|---|
| YAHPO | multifidelity/20/dev/yahpo/rbv2_ranger/40983/trainsize | 8 | 134 | 2 | 3 | 3 | 0 | trainsize | 0.03 | 1.00 |
| YAHPO | multifidelity/20/dev/yahpo/rbv2_ranger/41161/trainsize | 8 | 134 | 2 | 3 | 3 | 0 | trainsize | 0.03 | 1.00 |
| YAHPO | multifidelity/20/dev/yahpo/rbv2_xgboost/40499/trainsize | 14 | 170 | 10 | 2 | 2 | 0 | trainsize | 0.03 | 1.00 |
| YAHPO | multifidelity/20/dev/yahpo/rbv2_svm/24/trainsize | 6 | 118 | 3 | 1 | 2 | 0 | trainsize | 0.03 | 1.00 |
| YAHPO | multifidelity/20/dev/yahpo/rbv2_rpart/1220/repl | 5 | 110 | 1 | 3 | 1 | 0 | repl | 1.00 | 10.00 |
| YAHPO | multifidelity/20/dev/yahpo/rbv2_xgboost/375/trainsize | 14 | 170 | 10 | 2 | 2 | 0 | trainsize | 0.03 | 1.00 |
| YAHPO | multifidelity/20/dev/yahpo/rbv2_ranger/41161/repl | 8 | 134 | 2 | 3 | 3 | 0 | repl | 1.00 | 10.00 |
| HPOBench | multifidelity/20/dev/hpobench/mf/real/ml/rf/31/subsample | 4 | 100 | 1 | 3 | 0 | 0 | subsample | 0.10 | 1.00 |
| YAHPO | multifidelity/20/dev/yahpo/rbv2_svm/24/repl | 6 | 118 | 3 | 1 | 2 | 0 | repl | 1.00 | 10.00 |
| HPOBench | multifidelity/20/dev/hpobench/mf/real/ml/nn/146821/iter | 5 | 110 | 2 | 3 | 0 | 0 | iter | 3.00 | 243.00 |
| YAHPO | multifidelity/20/dev/yahpo/rbv2_rpart/41165/repl | 5 | 110 | 1 | 3 | 1 | 0 | repl | 1.00 | 10.00 |
| YAHPO | multifidelity/20/dev/yahpo/rbv2_aknn/1497/trainsize | 6 | 118 | 0 | 4 | 2 | 0 | trainsize | 0.03 | 1.00 |
| YAHPO | multifidelity/20/dev/yahpo/rbv2_svm/40975/repl | 6 | 118 | 3 | 1 | 2 | 0 | repl | 1.00 | 10.00 |
| YAHPO | multifidelity/20/dev/yahpo/rbv2_super/40984/repl | 38 | 267 | 18 | 13 | 7 | 0 | repl | 1.00 | 10.00 |
| MFPBench | multifidelity/20/dev/mfpbench/SO/mfh/mfh6_moderate | 6 | 118 | 6 | 0 | 0 | 0 | z | 1.00 | 100.00 |
| YAHPO | multifidelity/20/dev/yahpo/rbv2_glmnet/24/repl | 3 | 90 | 2 | 0 | 1 | 0 | repl | 1.00 | 10.00 |
| YAHPO | multifidelity/20/dev/yahpo/rbv2_ranger/41159/repl | 8 | 134 | 2 | 3 | 3 | 0 | repl | 1.00 | 10.00 |
| YAHPO | multifidelity/20/dev/yahpo/rbv2_xgboost/1476/repl | 14 | 170 | 10 | 2 | 2 | 0 | repl | 1.00 | 10.00 |
| YAHPO | multifidelity/20/dev/yahpo/iaml_glmnet/41146/trainsize | 2 | 77 | 2 | 0 | 0 | 0 | trainsize | 0.03 | 1.00 |
| YAHPO | multifidelity/20/dev/yahpo/rbv2_glmnet/334/repl | 3 | 90 | 2 | 0 | 1 | 0 | repl | 1.00 | 10.00 |

Table 14: Selected tasks ('multi-fidelity', 'test')

| benchmark_id | task | dimensions | n_trials | n_floats | n_integers | n_categoricals | n_ordinals | fidelity_type | min_budget | max_budget |
|---|---|---|---|---|---|---|---|---|---|---|
| YAHPO | multifidelity/20/test/yahpo/rbv2_rpart/38/repl | 5 | 110 | 1 | 3 | 1 | 0 | repl | 1.00 | 10.00 |
| YAHPO | multifidelity/20/test/yahpo/rbv2_xgboost/1480/repl | 14 | 170 | 10 | 2 | 2 | 0 | repl | 1.00 | 10.00 |
| YAHPO | multifidelity/20/test/yahpo/rbv2_xgboost/1476/trainsize | 14 | 170 | 10 | 2 | 2 | 0 | trainsize | 0.03 | 1.00 |
| YAHPO | multifidelity/20/test/yahpo/rbv2_super/40900/repl | 38 | 267 | 18 | 13 | 7 | 0 | repl | 1.00 | 10.00 |
| YAHPO | multifidelity/20/test/yahpo/rbv2_aknn/1476/trainsize | 6 | 118 | 0 | 4 | 2 | 0 | trainsize | 0.03 | 1.00 |
| YAHPO | multifidelity/20/test/yahpo/rbv2_super/458/trainsize | 38 | 267 | 18 | 13 | 7 | 0 | trainsize | 0.03 | 1.00 |
| YAHPO | multifidelity/20/test/yahpo/rbv2_aknn/50/trainsize | 6 | 118 | 0 | 4 | 2 | 0 | trainsize | 0.03 | 1.00 |
| YAHPO | multifidelity/20/test/yahpo/rbv2_super/41156/trainsize | 38 | 267 | 18 | 13 | 7 | 0 | trainsize | 0.03 | 1.00 |
| HPOBench | multifidelity/20/test/hpobench/mf/real/ml/xgboost/3/subsample | 4 | 100 | 3 | 1 | 0 | 0 | subsample | 0.10 | 1.00 |
| YAHPO | multifidelity/20/test/yahpo/rbv2_ranger/40923/trainsize | 8 | 134 | 2 | 3 | 3 | 0 | trainsize | 0.03 | 1.00 |
| HPOBench | multifidelity/20/test/hpobench/mf/real/ml/nn/146821/iter | 5 | 110 | 2 | 3 | 0 | 0 | iter | 3.00 | 243.00 |
| YAHPO | multifidelity/20/test/yahpo/rbv2_xgboost/41163/repl | 14 | 170 | 10 | 2 | 2 | 0 | repl | 1.00 | 10.00 |
| YAHPO | multifidelity/20/test/yahpo/rbv2_super/4154/repl | 38 | 267 | 18 | 13 | 7 | 0 | repl | 1.00 | 10.00 |
| YAHPO | multifidelity/20/test/yahpo/rbv2_super/458/repl | 38 | 267 | 18 | 13 | 7 | 0 | repl | 1.00 | 10.00 |
| YAHPO | multifidelity/20/test/yahpo/rbv2_svm/40981/trainsize | 6 | 118 | 3 | 1 | 2 | 0 | trainsize | 0.03 | 1.00 |
| YAHPO | multifidelity/20/test/yahpo/rbv2_aknn/40670/trainsize | 6 | 118 | 0 | 4 | 2 | 0 | trainsize | 0.03 | 1.00 |
| YAHPO | multifidelity/20/test/yahpo/rbv2_glmnet/41162/trainsize | 3 | 90 | 2 | 0 | 1 | 0 | trainsize | 0.03 | 1.00 |
| YAHPO | multifidelity/20/test/yahpo/rbv2_xgboost/40685/trainsize | 14 | 170 | 10 | 2 | 2 | 0 | trainsize | 0.03 | 1.00 |
| YAHPO | multifidelity/20/test/yahpo/rbv2_ranger/40923/repl | 8 | 134 | 2 | 3 | 3 | 0 | repl | 1.00 | 10.00 |
| YAHPO | multifidelity/20/test/yahpo/rbv2_aknn/1461/repl | 6 | 118 | 0 | 4 | 2 | 0 | repl | 1.00 | 10.00 |

Table 15: Selected tasks ('multi-objective', 'dev')

| benchmark_id | task | dimensions | n_trials | n_floats | n_integers | n_categoricals | n_ordinals | n_objectives |
|---|---|---|---|---|---|---|---|---|
| HPOBench | multiobjective/10/dev/hpobench/MO/tab/ml/nn/146821 | 5 | 110 | 0 | 0 | 0 | 5 | 2 |
| HPOBench | multiobjective/10/dev/hpobench/MO/tab/ml/xgboost/31 | 4 | 100 | 0 | 0 | 0 | 4 | 2 |
| Pymoo | multiobjective/10/dev/Pymoo/ManyO/unconstraint/wfg7_10_5 | 10 | 147 | 10 | 0 | 0 | 0 | 5 |
| YAHPO | multiobjective/10/dev/yahpo/mo/rbv2_xgboost/12/None | 14 | 170 | 10 | 2 | 2 | 0 | 2 |
| Pymoo | multiobjective/10/dev/Pymoo/ManyO/unconstraint/dtlz5 | 10 | 147 | 10 | 0 | 0 | 0 | 3 |
| HPOBench | multiobjective/10/dev/hpobench/MO/tab/ml/lr/14965 | 2 | 77 | 0 | 0 | 0 | 2 | 2 |
| HPOBench | multiobjective/10/dev/hpobench/MO/tab/ml/nn/9952 | 5 | 110 | 0 | 0 | 0 | 5 | 2 |
| YAHPO | multiobjective/10/dev/yahpo/mo/rbv2_xgboost/28/None | 14 | 170 | 10 | 2 | 2 | 0 | 2 |
| YAHPO | multiobjective/10/dev/yahpo/mo/rbv2_rpart/1476/None | 5 | 110 | 1 | 3 | 1 | 0 | 2 |
| HPOBench | multiobjective/10/dev/hpobench/MO/tab/ml/rf/12 | 4 | 100 | 0 | 0 | 0 | 4 | 2 |

Table 16: Selected tasks ('multi-objective', 'test')

| benchmark_id | task | dimensions | n_trials | n_floats | n_integers | n_categoricals | n_ordinals | n_objectives |
|---|---|---|---|---|---|---|---|---|
| Pymoo | multiobjective/10/test/Pymoo/MO/unconstraint/zdt1 | 30 | 240 | 30 | 0 | 0 | 0 | 2 |
| YAHPO | multiobjective/10/test/yahpo/mo/rbv2_xgboost/182/None | 14 | 170 | 10 | 2 | 2 | 0 | 2 |
| HPOBench | multiobjective/10/test/hpobench/MO/tab/ml/rf/168911 | 4 | 100 | 0 | 0 | 0 | 4 | 2 |
| HPOBench | multiobjective/10/test/hpobench/MO/tab/ml/xgboost/146212 | 4 | 100 | 0 | 0 | 0 | 4 | 2 |
| HPOBench | multiobjective/10/test/hpobench/MO/tab/ml/nn/3917 | 5 | 110 | 0 | 0 | 0 | 5 | 2 |
| YAHPO | multiobjective/10/test/yahpo/mo/lcbench/189873/None | 7 | 126 | 4 | 3 | 0 | 0 | 2 |
| HPOBench | multiobjective/10/test/hpobench/MO/tab/ml/lr/12 | 2 | 77 | 0 | 0 | 0 | 2 | 2 |
| HPOBench | multiobjective/10/test/hpobench/MO/tab/ml/lr/3 | 2 | 77 | 0 | 0 | 0 | 2 | 2 |
| HPOBench | multiobjective/10/test/hpobench/MO/tab/ml/rf/167119 | 4 | 100 | 0 | 0 | 0 | 4 | 2 |
| HPOBench | multiobjective/10/test/hpobench/MO/tab/ml/rf/167120 | 4 | 100 | 0 | 0 | 0 | 4 | 2 |

Table 17: Selected tasks ('multi-fidelity-objective', 'dev')

| benchmark_id | task | dimensions | n_trials | n_floats | n_integers | n_categoricals | n_ordinals | fidelity_type | min_budget | max_budget | n_objectives |
|---|---|---|---|---|---|---|---|---|---|---|---|
| YAHPO | momf/9/dev/yahpo/MOMF/repl/rbv2_xgboost/12/repl | 14 | 170 | 10 | 2 | 2 | 0 | repl | 1.00 | 10.00 | 2 |
| YAHPO | momf/9/dev/yahpo/MOMF/trainsize/rbv2_ranger/375/trainsize | 8 | 134 | 2 | 3 | 3 | 0 | trainsize | 0.03 | 1.00 | 2 |
| YAHPO | momf/9/dev/yahpo/MOMF/trainsize/iaml_ranger/1489/trainsize | 8 | 134 | 2 | 3 | 3 | 0 | trainsize | 0.03 | 1.00 | 3 |
| YAHPO | momf/9/dev/yahpo/MOMF/trainsize/rbv2_rpart/1476/trainsize | 5 | 110 | 1 | 3 | 1 | 0 | trainsize | 0.03 | 1.00 | 2 |
| YAHPO | momf/9/dev/yahpo/MOMF/trainsize/rbv2_ranger/6/trainsize | 8 | 134 | 2 | 3 | 3 | 0 | trainsize | 0.03 | 1.00 | 2 |
| YAHPO | momf/9/dev/yahpo/MOMF/trainsize/rbv2_xgboost/12/trainsize | 14 | 170 | 10 | 2 | 2 | 0 | trainsize | 0.03 | 1.00 | 2 |
| YAHPO | momf/9/dev/yahpo/MOMF/trainsize/rbv2_xgboost/28/trainsize | 14 | 170 | 10 | 2 | 2 | 0 | trainsize | 0.03 | 1.00 | 2 |
| YAHPO | momf/9/dev/yahpo/MOMF/trainsize/iaml_glmnet/1489/trainsize | 2 | 77 | 2 | 0 | 0 | 0 | trainsize | 0.03 | 1.00 | 2 |
| YAHPO | momf/9/dev/yahpo/MOMF/epoch/lcbench/167185/epoch | 7 | 126 | 4 | 3 | 0 | 0 | epoch | 1.00 | 52.00 | 2 |

Table 18: Selected tasks ('multi-fidelity-objective', 'test')

| benchmark_id | task | dimensions | n_trials | n_floats | n_integers | n_categoricals | n_ordinals | fidelity_type | min_budget | max_budget | n_objectives |
|---|---|---|---|---|---|---|---|---|---|---|---|
| YAHPO | momf/9/test/yahpo/MOMF/epoch/lcbench/189873/epoch | 7 | 126 | 4 | 3 | 0 | 0 | epoch | 1.00 | 52.00 | 2 |
| YAHPO | momf/9/test/yahpo/MOMF/epoch/lcbench/167152/epoch | 7 | 126 | 4 | 3 | 0 | 0 | epoch | 1.00 | 52.00 | 2 |
| YAHPO | momf/9/test/yahpo/MOMF/trainsize/iaml_xgboost/1489/trainsize | 13 | 165 | 10 | 2 | 1 | 0 | trainsize | 0.03 | 1.00 | 4 |
| YAHPO | momf/9/test/yahpo/MOMF/repl/rbv2_rpart/40499/repl | 5 | 110 | 1 | 3 | 1 | 0 | repl | 1.00 | 10.00 | 2 |
| YAHPO | momf/9/test/yahpo/MOMF/repl/rbv2_xgboost/182/repl | 14 | 170 | 10 | 2 | 2 | 0 | repl | 1.00 | 10.00 | 2 |
| YAHPO | momf/9/test/yahpo/MOMF/repl/rbv2_ranger/6/repl | 8 | 134 | 2 | 3 | 3 | 0 | repl | 1.00 | 10.00 | 2 |
| YAHPO | momf/9/test/yahpo/MOMF/trainsize/iaml_glmnet/1067/trainsize | 2 | 77 | 2 | 0 | 0 | 0 | trainsize | 0.03 | 1.00 | 2 |
| YAHPO | momf/9/test/yahpo/MOMF/trainsize/rbv2_xgboost/182/trainsize | 14 | 170 | 10 | 2 | 2 | 0 | trainsize | 0.03 | 1.00 | 2 |
| YAHPO | momf/9/test/yahpo/MOMF/trainsize/rbv2_ranger/40979/trainsize | 8 | 134 | 2 | 3 | 3 | 0 | trainsize | 0.03 | 1.00 | 2 |

# I   Experimental Results

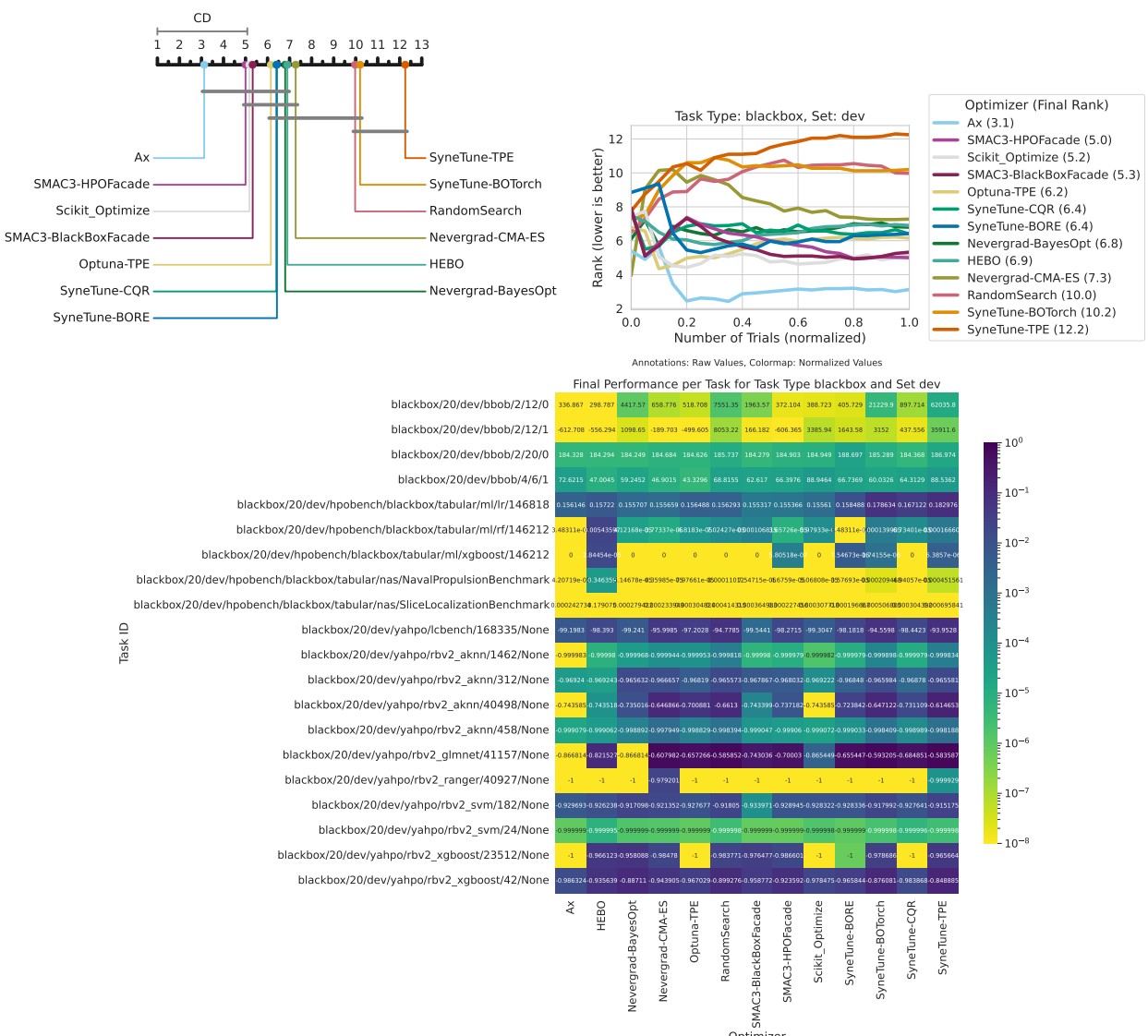

Figure 11: Result summary for Scenario blackbox and Set dev. First row: Critical difference diagram of final performance; rank over time based on statistical test. The grey area indicates non-significance based on statistical testing. Second row: Performance per problem. The annotations of the heatmap cells indicate the raw final performance (mean over seeds) and the colormap indicates the normalized values.

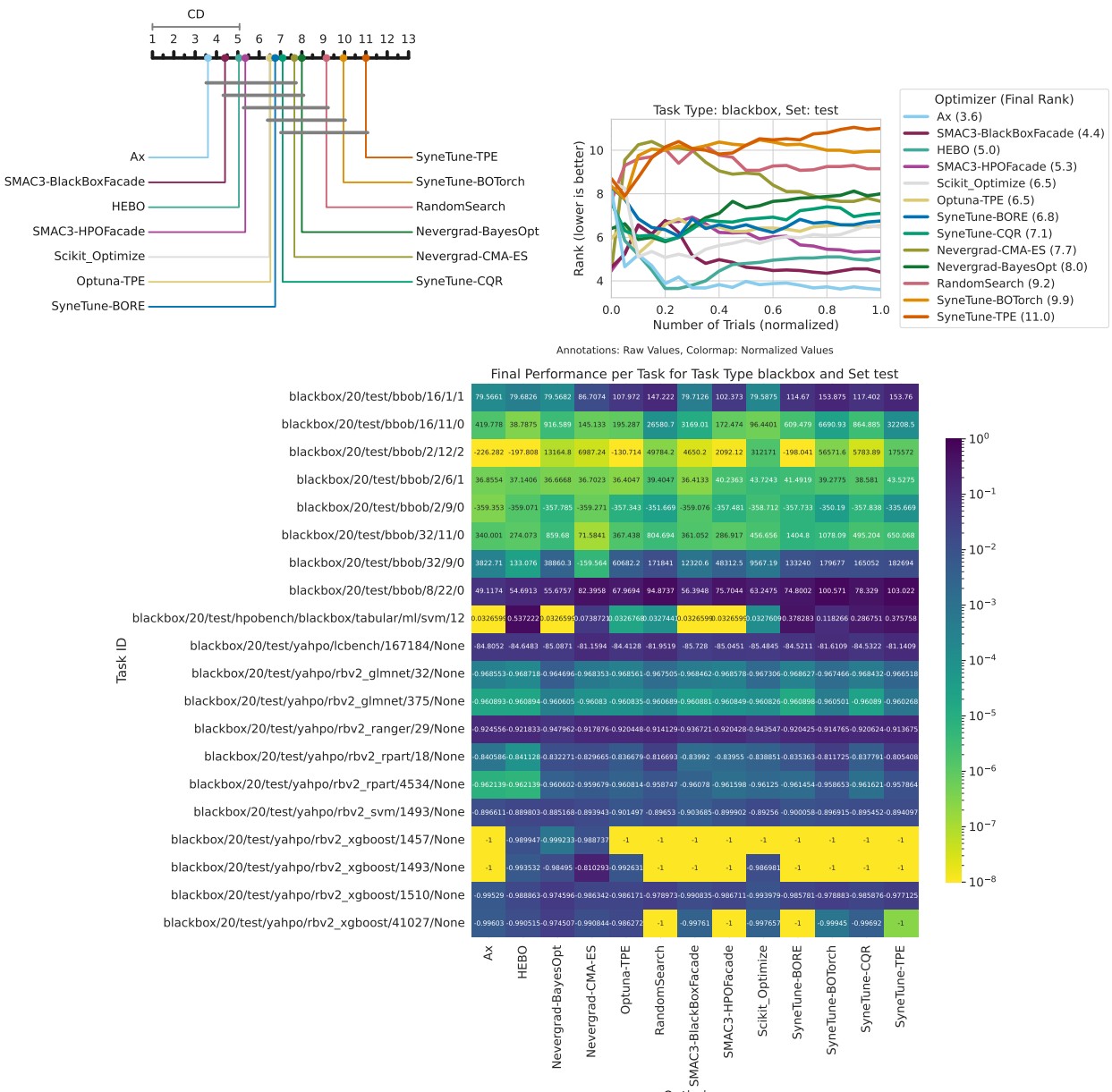

Figure 12: Result summary for Scenario blackbox and Set test. First row: Critical difference diagram of final performance; rank over time based on statistical test. The grey area indicates non-significance based on statistical testing. Second row: Performance per problem. The annotations of the heatmap cells indicate the raw final performance (mean over seeds) and the colormap indicates the normalized values.

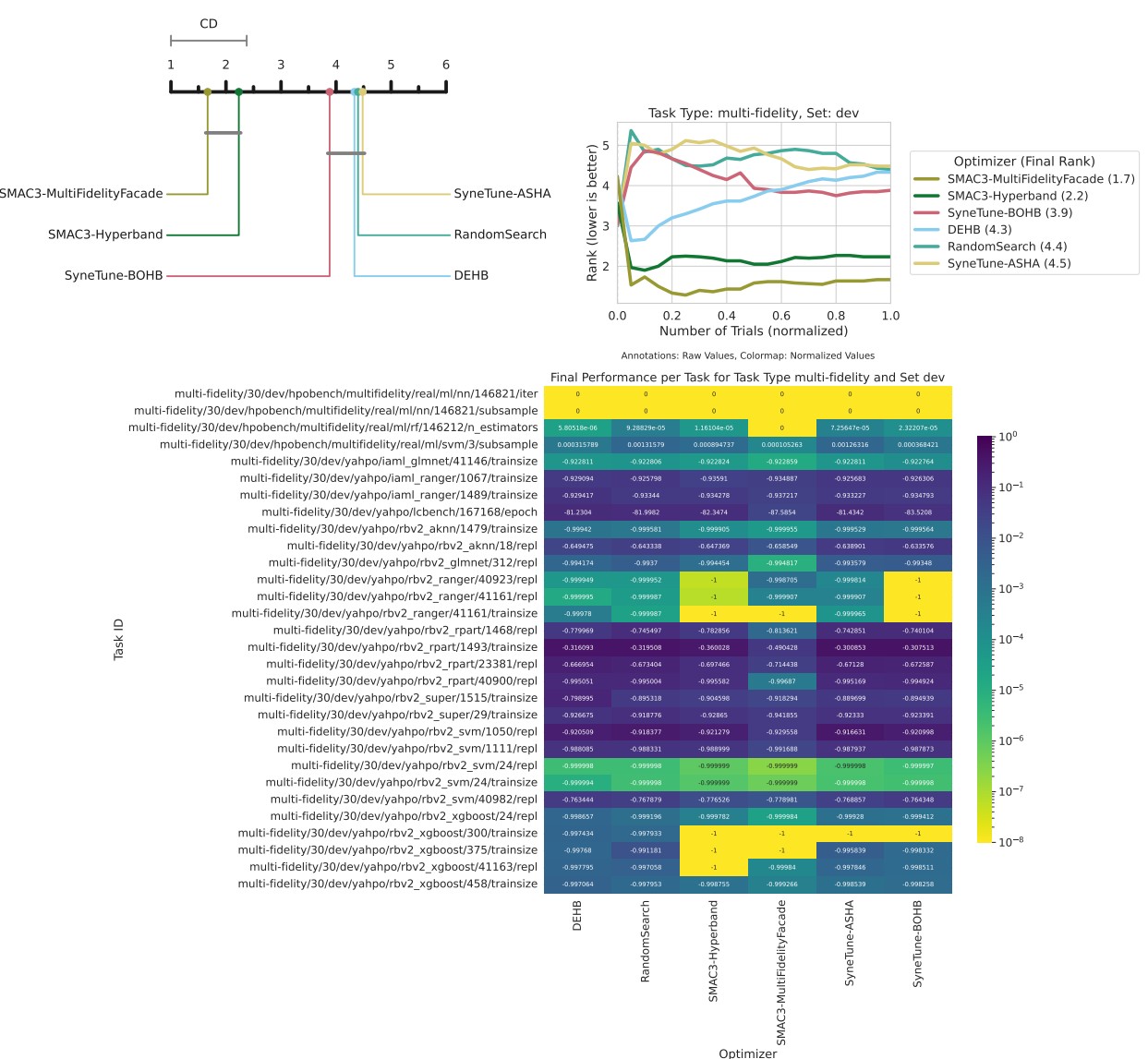

Figure 13: Result summary for Scenario multi-fidelity and Set dev. First row: Critical difference diagram of final performance; rank over time based on statistical test. The grey area indicates non-significance based on statistical testing. Second row: Performance per problem. The annotations of the heatmap cells indicate the raw final performance (mean over seeds) and the colormap indicates the normalized values.

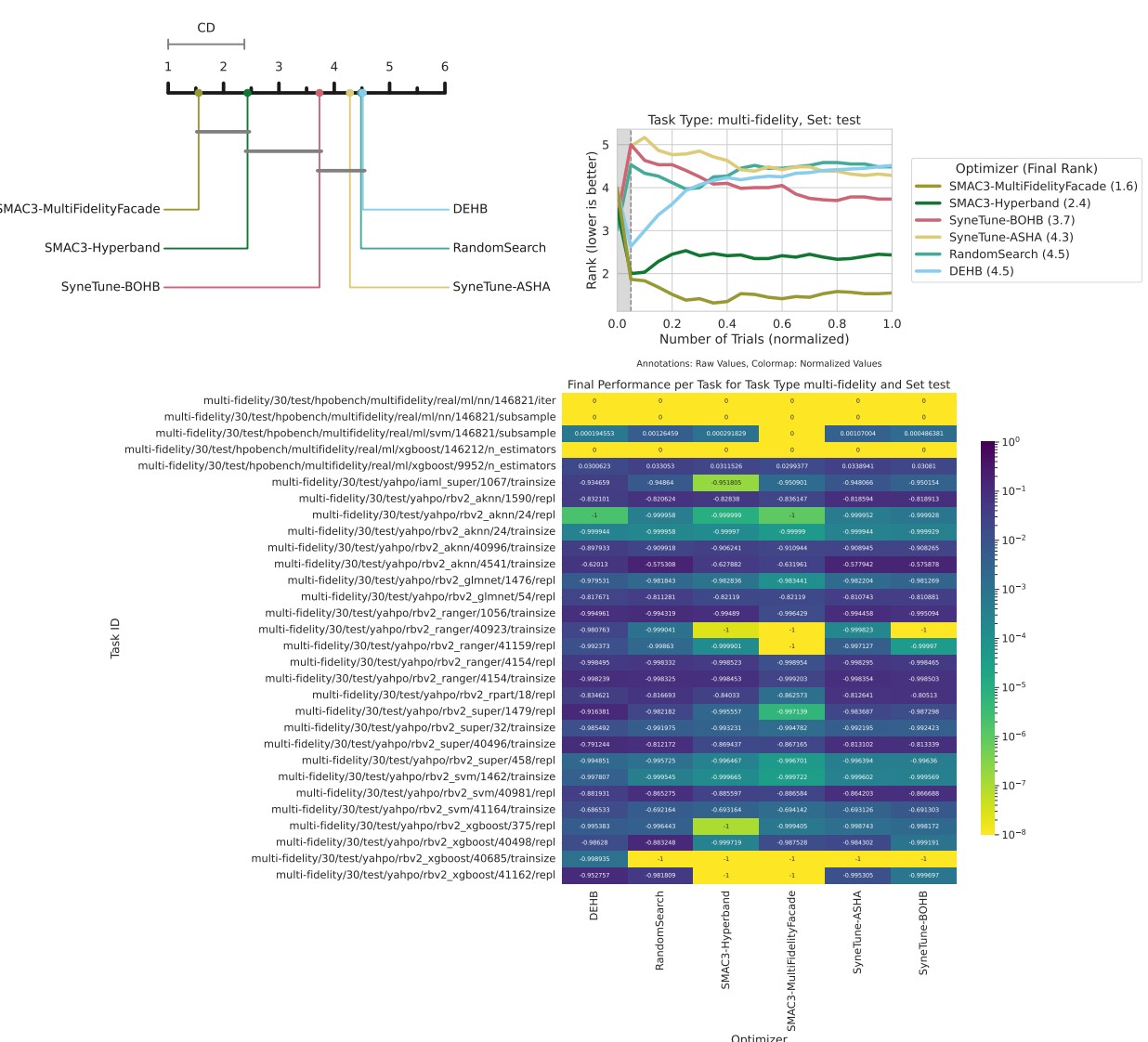

Figure 14: Result summary for Scenario multi-fidelity and Set test. First row: Critical difference diagram of final performance; rank over time based on statistical test. The grey area indicates non-significance based on statistical testing. Second row: Performance per problem. The annotations of the heatmap cells indicate the raw final performance (mean over seeds) and the colormap indicates the normalized values.

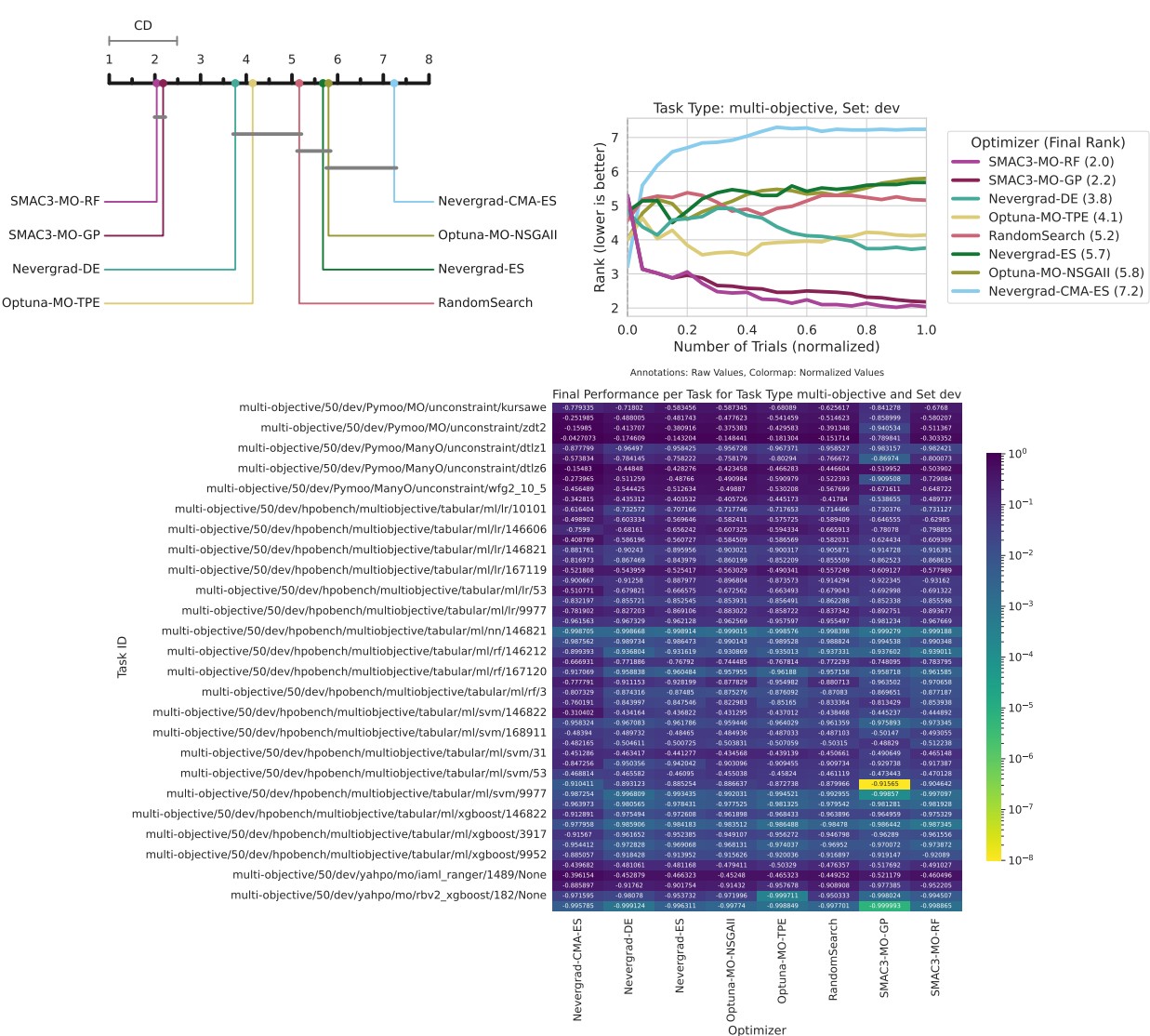

Figure 15: Result summary for Scenario multi-objective and Set dev. First row: Critical difference diagram of final performance; rank over time based on statistical test. The grey area indicates non-significance based on statistical testing. Second row: Performance per problem. The annotations of the heatmap cells indicate the raw final performance (mean over seeds) and the colormap indicates the normalized values.

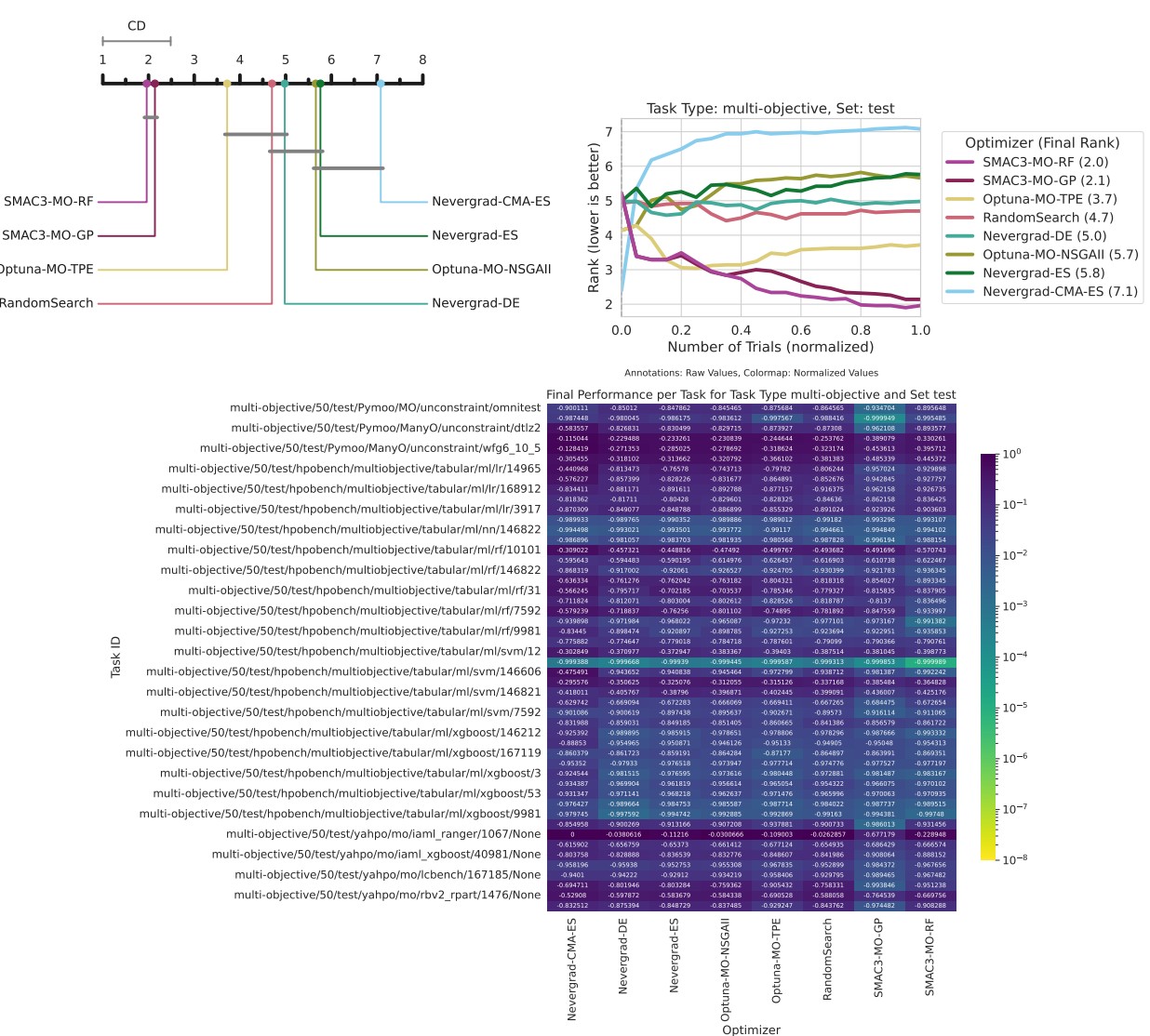

Figure 16: Result summary for Scenario multi-objective and Set test. First row: Critical difference diagram of final performance; rank over time based on statistical test. The grey area indicates non-significance based on statistical testing. Second row: Performance per problem. The annotations of the heatmap cells indicate the raw final performance (mean over seeds) and the colormap indicates the normalized values.

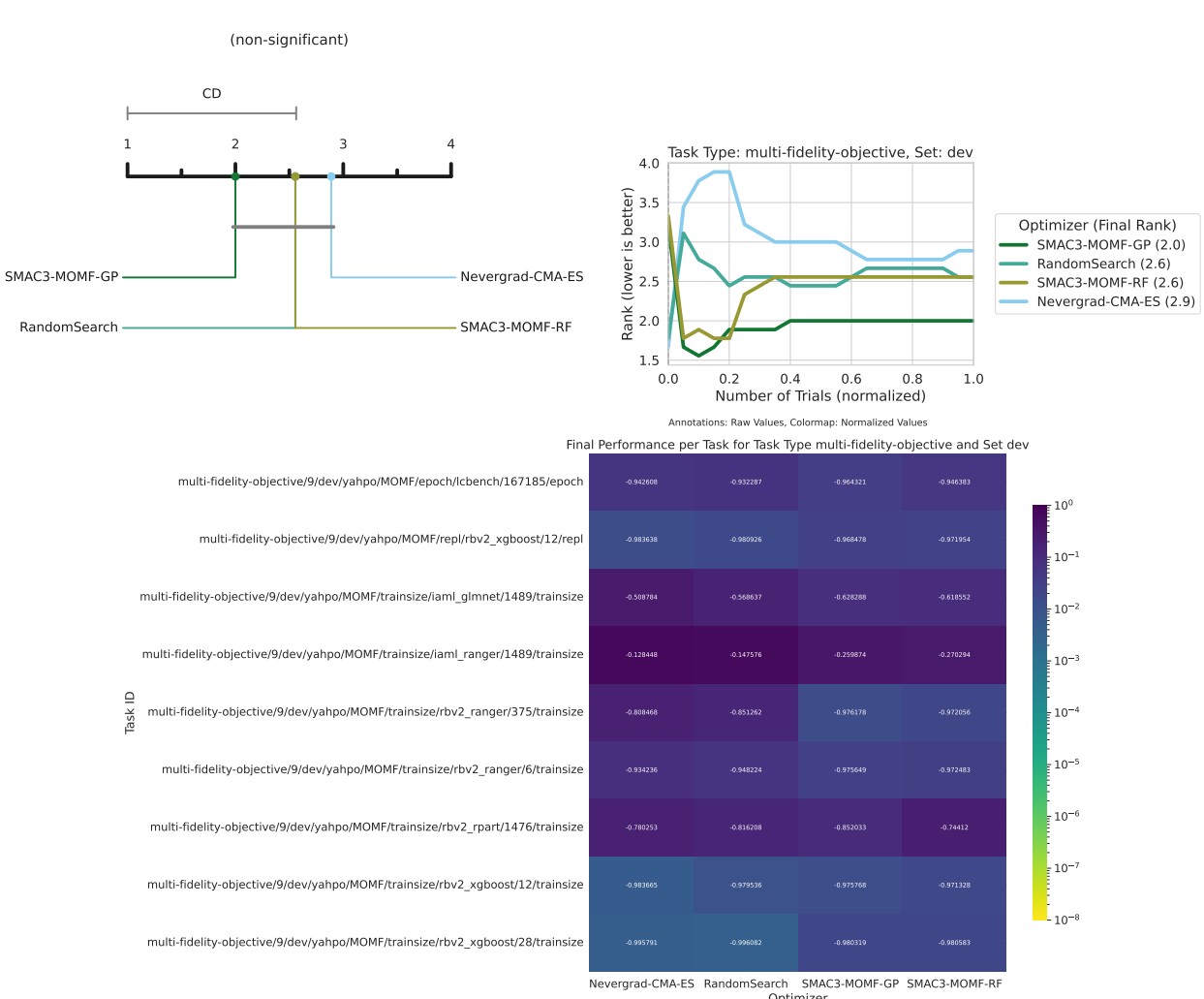

Figure 17: Result summary for Scenario multi-fidelity-objective and Set dev. First row: Critical difference diagram of final performance; rank over time based on statistical test. The grey area indicates non-significance based on statistical testing. Second row: Performance per problem. The annotations of the heatmap cells indicate the raw final performance (mean over seeds) and the colormap indicates the normalized values.

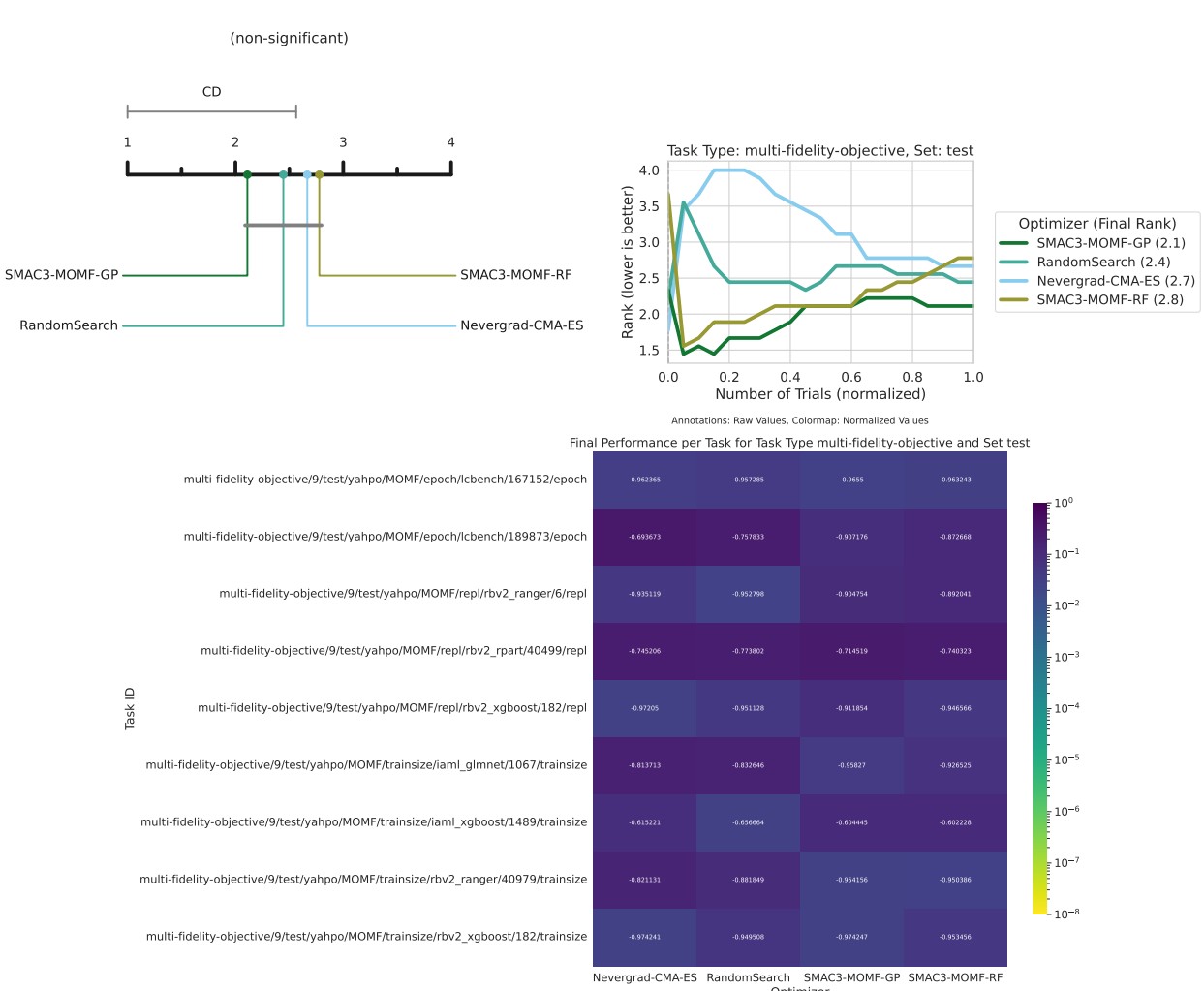

Figure 18: Result summary for Scenario multi-fidelity-objective and Set test. First row: Critical difference diagram of final performance; rank over time based on statistical test. The grey area indicates non-significance based on statistical testing. Second row: Performance per problem. The annotations of the heatmap cells indicate the raw final performance (mean over seeds) and the colormap indicates the normalized values.

