# OpenReview forum: "carps: A Framework for Comparing N Hyperparameter Optimizers on M Benchmarks"
_TMLR — Decision pending for TMLR_

### Review · Reviewer_HeNc · 2025-08-26

**Summary Of Contributions:**

The paper proposes a comprehensive and efficient benchmarking framework for hyperparameter optimization. Existing benchmarking datasets have been published in the literature; the proposed framework, introduced as CARPS, combines them. Importantly, evaluating N optimizers on M benchmarks typically requires a significant amount of computation; however, the CARPS circumvents this by utilizing a subset selection method based on the star discrepancy.

**Additional Comments:**

Consider explaining Critical Difference Diagrams in the Figure caption for readers who may be only skimming the text.

**Audience:**

Yes

**Audience Explanation:**

I would expect a broad interest from the TMLR audience, as the proposed methodology of the paper may be generally applicable for large-scale evaluations.

**Claims And Evidence:**

Yes

**Claims Explanation:**

The paper provides clear statistics and information on the benchmarks used, along with precise evaluations in the figures. The text is easy and pleasant to read for me, even though I do not usually read papers on hyperparameter optimization.

**Requested Changes:**

I have some suggestions on the following. I am not familiar with the terminology of hyperparameter optimization, so I would appreciate more detailed explanations in the captions to clarify the abbreviated keywords used in the Figures.

- Figure 3, HP Types, x-axis: I am more familiar with the float and int types (e.g., a ridge regularization parameter is of type float and the number of epochs is of type int). What would be a cat type and ord type HP? What does even ord type refer to?

- Figure 4: What does the 3-dimensional optimizer performance vector refer to?

- Eq 4. are $ y_i $ constrained to be in the interval $[0, 1)$ ? If not, please adjust the sizes of the encapsulating box with $y_{\text{max}}$ and $y_{\text{min}}$ values.

- Eq 5. subscript of sup should be $ q \in [0, 1)$ -- that is, remove $d$ -- and that superscript $d$ should go to the denominator of the ratio and above the interval $[0, q)$. The current formulation is mathematically illegal.

---

### Review · Reviewer_njJX · 2025-09-30

**Summary Of Contributions:**

This paper proposes Comprehensive Automated Research Performance Studies (carps), a benchmarking framework for hyperparamter optimization (HPO).  Specifically, this is primarily a software/benchmark contribution spanning a vast number of tasks and types of HPO problems, e.g., blackbox (BB), multi-fidelity (MF), multi-objective (MO), and multi-fidelity-multi-objective (MOMF).

### Strengths
I have a very favorable opinion of the paper, given the following strengths:
- **Scope**: Overall, I see this work as a substantial contribution to the HPO community.  The reason for this is due to the number of tasks, focus on multiple types of HPO tasks, unified API, accessibility through various forms of parallelism, and the end goals of improved benchmarking and reproducibility.
- **Code & Documentation**: The repository is very well documented and clearly structured.  Briefly comparing to other benchmarking/evaluation repositories I've used in the past, this is arguably one of the most well-documented.  There is also documentation on how to create extensions (new optimizers/ benchmarks), which is great to see.

### Clarifications
Generally, I have a favorable opinion of the paper, so I have titled this section Clarifications rather than Weaknesses.  These are primarily focused on the criteria for subset selection.
- **Selection Metric**: The authors motivate star discrepancy and validate that subsets preserve optimizer rankings compared to the complete set, which is excellent. However, there’s little discussion of alternative selection metrics (e.g., clustering). Including such a discussion would strengthen the case that star discrepancy is the right choice.

**Additional Comments:**

Hyperparameter optimization is not my core area of expertise, though I have some familiarity with the topic.

**Audience:**

Yes

**Audience Explanation:**

Yes, this is definitely of interest.  HPO is a well-established research area that intersects with ML, among other disciplines, so given this is the most comprehensive benchmarking suite, I'd argue it is of notable interest to the TMLR community.

**Broader Impact Concerns:**

None.

**Claims And Evidence:**

Yes

**Claims Explanation:**

The paper proposes a benchmarking suite for HPO and delivers on this with a relatively easy-to-use, well-documented, and extensive (in terms of both optimizations, benchmarks, and tasks).

**Requested Changes:**

I would like some clarification on the choice of star discrepancy compared to other similarity measures over the performance space.

---

### Review · Reviewer_fmVf · 2026-02-13

**Summary Of Contributions:**

This paper introduces carps, a comprehensive benchmarking framework for Hyperparameter Optimization (HPO). Its main contributions are:

(1) a complete experimentation pipeline with a unified API for integrating optimizers and tasks;

(2) access to 3336 HPO tasks from 5 benchmark collections and 26 optimizer variants;

(3) a novel method for sub-selecting representative benchmark tasks using star discrepancy minimization, yielding computationally feasible development/test sets.

- Key Strengths: Practical utility, impressive scale and diversity of tasks/optimizers, and principled subset selection method.

- Key Weaknesses: Subset selection depends on a performance space defined by only three optimizers may introduce bias; validation of subsets could be stronger; baseline result analysis is somewhat shallow.

**Audience:**

Yes

**Audience Explanation:**

The paper is highly relevant to researchers developing HPO algorithms, AutoML systems, and those interested in benchmarking and meta-learning. It lowers the barrier to rigorous empirical work and promotes standardized evaluation.

**Claims And Evidence:**

Yes

**Claims Explanation:**

The framework's functionality and scale are clearly demonstrated. The subset selection methodology is well-reasoned, though its claim of "representativeness" could use stronger empirical validation.

**Requested Changes:**

- Validate Subset Representativeness: Add an experiment evaluating a held-out optimizer (not used in subset creation) on both the full task set and the proposed test subset. This is to show that performance rankings are correlated, demonstrating the subset preserves task landscape structure.

- Discussing the potential bias from using **performance space defined by only three optimizers** to define the task space. Is there any methods to incorporate more features to address this issue？(better feature space and better optimizer selection)

- Clarify MO Metrics: Explicitly state how multi-objective performance was aggregated for ranking in Section 8 and figure captions.

- The repository was expired. Can you renew the code link?

---

> ### Author Response · Authors · 2026-02-27
> **Thank you reviewer fmVf for your review! We addressed your points regarding subset representativeness, performance space bias, and MO clarification in the comment.**
>
> Dear reviewer fmVf,
>
> Thank you for your review.
> In the following, we will address your requested changes. All changes made to the paper based on your review are marked with magenta font color.
>
> First of all, we apologize for the expired repository link; it was set to expire 6 months after submission, which was our assumed turnaround time for TMLR. It has now been updated, and the repository is available again.
>
> **Subset Representativeness**
> Following your suggestion, we validated the subset’s representativeness by running an additional optimizer (Ax) and confirmed that the ranking remains the same.
> The space for subselect selection is spanned by the performances of three common algorithm classes, namely random search, evolutionary algorithms, and Bayesian optimization (BO). For validation, we select the optimizer Ax, which is a framework for BO, and thus falls into the category of being a representative algorithm.
> We have added this validation to the paper in Section 7.2.
>
> **Bias from Performance Space spanned by Three Optimizers**
>
> We can distinguish between bias introduced by the choice of optimizers and bias introduced by the choice of features.
>
> For the former, we select three representative optimizers, which are distinct in their algorithm and thus optimization behavior, namely, we select random search, BO, and CMA-ES (evolutionary algorithm). In general, one could further extend this set of optimizers to other algorithm classes (e.g., local search algorithms or approximative gradient optimizers); however, both are rarely used for HPO. To showcase the impact of our subset selection approach, we limit ourselves to the commonly used algorithm classes mentioned above, but we could easily extend this representative subset in the future if new algorithm classes become more widely used. We acknowledge that this introduces a bias that could make it harder for new algorithm classes to demonstrate their performance gains. We added this discussion in Section 7.2.
>
> We argue that directly using optimizer performance is the optimal representation for selecting subsets. The performance, in fact, is the target, and it directly reflects the behavior of optimizers and characterizes them. Any other (meta or instance) features about the task or the optimizer would act as a proxy for performance and would then introduce bias for the selection process. In addition, as noted in the paper, recent work suggests that state-of-the-art features for hyperparameter optimization and blackbox tasks do not capture the objective function structure well enough for AutoML approaches \[Nikolikj et al., 2023; Long et al., 2023; Vermetten et al., 2023\]. The representation of tasks in the performance space has also been shown to be effective for subselection and for algorithm selection \[Cenikj et al., 2023; Benjamins et al., 2024; Cenikj et al., 2025\].
> We note that meta or instance features are commonly used for either saving compute time (e.g., as part of online algorithm selection) or to gain a deeper understanding of the benchmark instances. Neither is the primary objective for our subset selection.
>
> **MO Metrics**
> Regarding the calculation of the MO metric, we utilize the hypervolume.
> We first calculate the reference point per task across all collected runs as the maximum of each objective. Then, for each run, we calculate the hypervolume iteratively using all tuples of observed objective values. We added this to the paper under 8.1.
>
> Best wishes
> The authors
>
> |  |  |
> | :---- | :---- |
> | \[Benjamins et al., 2024\] | https://dl.acm.org/doi/abs/10.1145/3638530.3654291 |
> | \[Cenikj et al., 2023\] | https://dl.acm.org/doi/abs/10.1145/3583131.3590401 |
> | \[Cenikj et al., 2025\] | https://direct.mit.edu/evco/article/doi/10.1162/evco\_a\_00370/128316 |
> | \[Long et al., 2023\] | https://link.springer.com/chapter/10.1007/978-3-031-30229-9\_25 |
> | \[Nikolikj et al., 2023\] | https://ieeexplore.ieee.org/abstract/document/10254146/ |
> | \[Vermetten et al., 2023\] | https://proceedings.mlr.press/v224/vermetten23a.html |

---

### Decision · Action_Editor_1yAF · 2026-06-13

**Recommendation:** Accept as is

**Additional Comments:**

The reviewers were unanimous in their support for the paper, praising its high quality, usefulness, and reproducibility via a clean codebase and simple interface. The subset selection via star discrepancy also makes computation far less burdensome.

**Audience:**

Yes

**Audience Explanation:**

HPO is a large community addressing a critical problem in machine learning. A benchmark of this size, scope, and practicality (through subset selection), with clean code, would certainly be of interest.

**Claims And Evidence:**

Yes

**Claims Explanation:**

Yes, the paper offers a benchmark with a unified and lightweight HPO interface, with explicit callouts from the reviewers on how clean it is.

The paper claims that the subset selection methodology yields representative subsets that preserve optimizer rankings, and this is verified by using a held-out optimizer and showing consistent rankings across splits.